# Symbolic Autoencoding
# With Straight-Through Gradient Estimators

## Abstract

Self-supervised autoregressive models have achieved significant success across diverse domains, including text, audio, and biological sequences. However, these models often rely heavily on large samples of aligned (parallel) data, limiting their applicability in low-resource settings. To address this limitation, we propose Symbolic Autoencoding ($\Sigma$AE) with Straight-Through Gradient Estimators (STGEs)—a latent variable model where the latent space consists of sequences of categorical random variables, resembling sentences in an emergent symbolic language. $\Sigma$AE is trained end-to-end using a family of straight-through gradient estimators. In the unsupervised mode, $\Sigma$AE learns to compress input data into symbolic sentences and reconstructs the data from this emergent language. In weakly supervised settings, $\Sigma$AE further grounds the latent language by leveraging supervised training on the small amount of parallel data available. We evaluate $\Sigma$AE with three well-known quantization mechanisms on four text sequence transduction tasks. Our results show that $\Sigma$AE outperforms baseline methods, particularly in low-resource scenarios with limited parallel data.

## 1 Introduction and Preliminaries

Mapping data between symbolic systems is a fundamental problem in information theory and machine learning Kaiser & Bengio (2018); Baziotis et al. (2019); Fortuin et al. (2019). Whether converting natural language into structured representations or relating biological sequences to their functions, the central challenge lies in effectively encoding and representing information from one symbolic system into another. In natural language processing, for example, this may involve translating sentences into semantic graphs, requiring both syntactic parsing and an understanding of deeper semantic structures. Sánchez et al. (2023)

In cases where large-scale parallel datasets are available—such as aligned language pairs in machine translation—supervised models can be trained to perform these mappings directly. However, in the more common scenario where parallel data is scarce or unavailable, how can we still learn to map sequences between two symbolic systems? Magueresse et al. (2020); Lample & Conneau (2019); Joshi et al. (2020); Gregor et al. (2013).

To address this challenge, we propose Symbolic Autoencoding ($\Sigma$AE), a weakly-supervised framework for sequence transduction across symbolic systems $X$ and $Z$. $\Sigma$AE is designed to leverage both non-parallel data and limited amounts of parallel data to establish bidirectional mappings between two systems. At its core, $\Sigma$AE employs a latent variable model where the latent space comprises sequences of categorical random variables—discrete symbolic representations with variable lengths. These latent sequences serve as an emergent language, enabling both lossy compression (via autoencoding) and alignment with parallel data via standard supervised training when available.

The training process alternates between two complementary tasks:

- **Unsupervised Autoencoding:** $\Sigma$AE reconstructs sequences through bidirectional mappings:
    - $Z \xrightarrow{M_{zx}} X \xrightarrow{M_{xz}} Z$, where $M_{zx}$ serves as the encoder and $M_{xz}$ as the decoder; or
    - $X \xrightarrow{M_{xz}} Z \xrightarrow{M_{zx}} X$, where $M_{xz}$ acts as the encoder and $M_{zx}$ as the decoder.
    Both mappings utilize discrete bottlenecks to learn emergent symbolic representations.

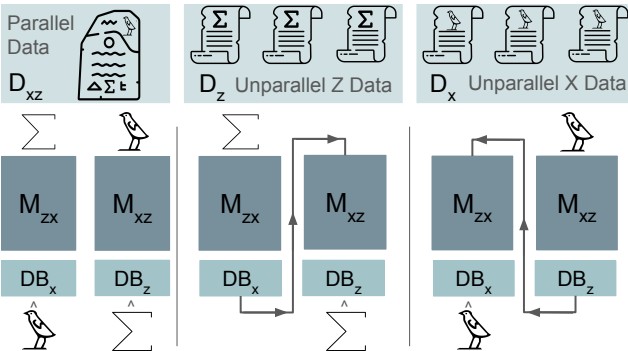

Figure 1: Illustration of the abstract flow of data in the symbolic autoencoding ($\Sigma$AE) framework, exemplified with the Rosetta Stone problem. Two sequence-to-sequence models ($M_{xz}$ and $M_{zx}$) are trained with both parallel data (the Rosetta Stone) through next-token prediction and unparallel data through connecting the models with a discrete bottleneck layer ($DB_x$ and $DB_z$) to autoencode each language using the other as its hidden representation.

- **Weakly-Supervised Refinement:** When paired data is available, supervised training is employed to refine the alignment between latent representations and the ground-truth mappings. This is achieved by minimizing the cross-entropy loss on each of the model outputs.

Key to $\Sigma$AE is the discrete bottleneck layer, which outputs both token probability distributions and quantized samples, enabling end-to-end differentiability through straight-through gradient estimators Bengio et al. (2013) and the reparameterization trick. We explore three quantization mechanisms—softmax-argmax, Gumbel-softmax-argmax, and vector quantization (VQ-VAE)—to optimize training efficiency and stability. To mitigate hidden sequence collapse, where EOS tokens are prematurely selected and the latent sequences are ignored Bowman et al. (2016); Sánchez et al. (2023); Zhao et al. (2018), we introduce a gradient estimator for EOS masking probabilities, ensuring gradients are propagated across the full sequence.

In the absence of parallel data, $\Sigma$AE learns an emergent symbolic representation within the bottleneck, which can be interpreted as a lossy compression mechanism. This emergent language effectively captures the underlying structure of the input symbolic system.

We evaluate $\Sigma$AE on four sequence transduction benchmarks (SCAN, PCFG SET, CFQ, and COGS) and demonstrate superior performance in both unsupervised and weakly-supervised settings. Our results highlight the framework's ability to learn compressed symbolic representations, enabling generalization across modalities and establishing $\Sigma$AE as a robust approach for symbolic sequence transduction.

**Related work.**

Our work intersects with several key areas in unsupervised and weakly supervised learning through discrete representations. Baziotis et al. (2019) connected two encoder-decoder models via a hidden sequence layer, employing a reconstruction loss and a language model prior loss for unsupervised text compression. Kaiser & Bengio (2018) explored semantic hashing (Salakhutdinov & Hinton, 2009) and the Gumbel-Softmax trick (Jang et al., 2017) for generating interpretable, discrete encodings. Kaiser et al. (2018) further experimented with the Gumbel-Softmax trick, semantic hashing, and two vector quantization variants van den Oord et al. (2017) to learn a compressed discrete latent variable for fast parallel decoding in a seq2seq model. Sánchez et al. (2023) trained Hidden Schema Networks, a discrete VAE model trained via the Gumbel-Softmax trick mapping input sequences to random walk instances on a learned graph, and demonstrated the model's ability to learn interpretable representations of structured data. Similarly, Fortuin et al. (2019) investigated training VAEs with discrete bottlenecks and examined the use of continuous paths alongside discrete ones. Zhao et al. (2018) proposed a fixed-length discrete latent variable model for learning interpretable dialogue action representations via Gumbel-Softmax and maximizing the mutual information between data and latent actions.

In the realm of semi-supervised learning, Kingma et al. (2014) studied semi-supervised autoencoding by treating the classification label as a categorical hidden variable, emphasizing the integration of labeled and unlabeled data.

Zhu et al. (2017) and He et al. (2016) enforced consistency across translation tasks, with Zhu et al. (2017) using adversarial networks and He et al. (2016) employing Reinforcement Learning (RL) to update the models. Using REINFORCE Williams (1992) to optimize the discrete latent variables has been explored in other contexts. Evtimova et al. (2018) investigated emergent communication, where two agents collaboratively converse about images and textual descriptions using sequences of binary symbols from a fixed set to match the images and text. Similarly, Miao & Blunsom (2016) proposed a generative variational auto-encoding sentence compression model that compresses a document into a summary sentence and reconstructs the document from the summary, employing a latent language model as the encoder. Our work introduces a parallel approach, in contrast with RL and cycle-consistency based methods, leveraging straight-through gradient estimators to train symbolic probabilistic generative models end-to-end.

Furthermore, numerous studies have focused on the discretization of elements and representations in neural networks Liu et al. (2022; 2021); Tamkin et al. (2023); Peng et al. (2018); Maddison et al. (2016).

Our proposed solution also parallels the technique of back-translation (Sennrich et al., 2015; Çaglar Gülçehre et al., 2015; 2017), which typically involves training an intermediate system on parallel data to translate target monolingual data into the source language, thereby generating synthetic parallel corpora for further training Edunov et al. (2018). The $\Sigma$AE framework is akin to an online version of back-translation, where the intermediate system is continuously improving without storing synthetic sequences.

## 2 $\Sigma$AE FRAMEWORK

To mathematically formalize the symbolic latent variable learning process, we will first describe the training of a single discrete random variable. We will then extend this framework to the case of learning sequences of discrete random variables.

### 2.1 BACKGROUND: TRAINING A SINGLE DISCRETE RANDOM VARIABLE

Consider a probabilistic latent variable model that maps an input $x$ to a probability distribution over a latent variable $z$, denoted as $P_\theta(Z|X = x)$. We assume the ability to sample from this distribution and compute a loss $l(z)$ for a sample, which may be influenced by downstream components in the computation graph.

Even for continuous random variables, optimizing this process is challenging, as the loss is stochastic and non-differentiable. The average loss for a given input is expressed as $\mathbb{E}_{z \sim p_\theta(Z|X=x)}[l(z)]$, and the overall loss across the input distribution is $\mathbb{E}_{x \sim p(x)}[\mathbb{E}_{z \sim p_\theta(Z|X=x)}[l(z)]]$.

### 2.1.1 OPTIMIZATION METHODS

Several methods have been proposed to minimize this average loss:

- **REINFORCE** Williams (1992) uses the identity:

$$\nabla_\theta \mathbb{E}_{z \sim p_\theta(Z|X=x)}[l(z)] = \mathbb{E}_{z \sim p_\theta(Z|X=x)}[l(z)\nabla_\theta \log p_\theta(Z|X = x)] \tag{1}$$

This method estimates the gradient for the average loss using a one-sample Monte Carlo approximation:

$$\nabla_\theta \mathbb{E}_{z \sim p_\theta(Z|X=x)}[l(z)] \approx l(z)\nabla_\theta \log p_\theta(Z|X = x_i) \tag{2}$$

Notably, the gradient is estimated directly from the distribution samples, without requiring differentiability of the distribution. Variants such as REINFORCE with baselines were later introduced to reduce gradient variance Gu et al. (2015).

- **Continuous averaging and annealing Methods** methods do not sample from the distribution; instead, they compute a deterministic weighted sum of the embeddings. Additionally, Graves (2016) introduces a halting mechanism for variable-length, adaptive computation using continuous embeddings rather than discrete representations.
  In Correia et al. (2019); Peters et al. (2019) gradual temperature annealing is employed to make the distribution increasingly deterministic. At each training step, the objective is minimizing $l(\mathbb{E}_{z \sim p_\theta(Z|X=x)} z)$—loss of the average embedding- rather than minimizing $l(z)$ for specific samples $z \sim p_\theta(Z|X=x)$. This expectation is differentiable, allowing gradient-based optimization. Over the course of training, the temperature is annealed to zero, transitioning the expected value to the argmax of the distribution, effectively producing discrete samples.
- **Reparameterization trick** Kingma & Welling (2013) offers a more stable alternative. It expresses the random variable as a differentiable, deterministic function of noise $\epsilon \sim \mathcal{D}_\epsilon$ and distribution parameters $\theta$: $z = g_\theta(\epsilon, x)$. For instance, sampling from a Gaussian distribution $z \sim \mathcal{N}(\mu, \sigma^2)$ that can be reparameterized as $z = \mu + \sigma\epsilon$ where $\epsilon \sim \mathcal{N}(0, 1)$. This allows the gradient of the expected loss to be written as:

$$\nabla_\theta \mathbb{E}_{z \sim p_\theta(Z|X=x)}[l(z)] = \nabla_\theta \mathbb{E}_{\epsilon \sim \mathcal{D}_\epsilon}[l(g_\theta(\epsilon, x))] = \mathbb{E}_{\epsilon \sim \mathcal{D}_\epsilon}[\nabla_\theta l(g_\theta(\epsilon, x))]. \tag{3}$$

This can be efficiently approximated using a one-sample Monte Carlo estimate, enabling gradient-based optimization Kingma & Welling (2013); Gregor et al. (2013).

### 2.1.2 GENERALIZATION OF REPARAMETERIZATION TRICK TO CATEGORICAL RANDOM VARIABLES

Our goal is to learn symbolic representations, which are sequences of discrete tokens. These tokens follow a categorical distribution $p_\theta(Z|X=x)$, where $Z$ represents a token.

For categorical random variables, the reparameterization trick can be applied using the Gumbel-Softmax distribution Jang et al. (2017); Maddison et al. (2016). Given a probability vector $p_\theta(x) = P_\theta(Z|X=x)$ for the categorical distribution, the Gumbel-Softmax reparameterization enables sampling from $p_\theta$ via:

$$z = \text{one-hot}[\arg\max_i p_g(p_\theta(x), \tau_g)], \tag{4}$$

where $p_g(p_\theta(x), \tau_g) = \text{softmax}(\frac{\log p_\theta(x) + g}{\tau_g})$ and $g \sim G(0, 1)$ is is sampled from the Gumbel distribution.

While this reparameterization captures the sampling process, the $\arg\max$ function is non-differentiable. To circumvent this, the straight-through estimator replaces the gradient of the $\arg\max$ with an identity function, allowing gradient flow through the non-differentiable operation Bengio et al. (2013).

### 2.1.3 GRADIENT APPROXIMATION FOR DISCRETE VARIABLES

More formally, given a sampled $z = \text{one-hot}[\arg\max_i p_g(p_\theta(x), \tau_g)]$, the gradient of the loss with respect to model parameters can be approximated as:

$$\nabla_\theta l(z) \approx \nabla_z l(z) \nabla_\theta z = \nabla_z l(z) \nabla_\theta \left[ \text{one-hot}[\arg\max_i p_g(p_\theta(x), \tau)] \right] \approx \nabla_z l(z) \nabla_\theta p_g(p_\theta(x), \tau_g)$$
$$\tag{5}$$

This approach copies the gradient from the output of the $\arg\max$ to its input, enabling backpropagation through the parameters. In automatic differentiation frameworks, this is often implemented as: $z \leftarrow z + p_g - \text{sg}(p_g)$ where the **sg** is the stop-gradient operator, ensuring that gradients are not propagated back through the corresponding computation subgraph van den Oord et al. (2017).

### 2.2 DISCRETE BOTTLENECK

We refer to the quantization layer in the model as the **Discrete Bottleneck (DB)**. In this framework, sampling from the distribution is required to propagate information forward in the computation graph. Importantly, to maintain the discreteness of the latent sequence elements, continuous averaging over

embeddings (as discussed in section 2.1.1) is not permitted. Allowing such averaging would introduce information leakage through the continuous averaging weights, enabling the encoder to indirectly pass continuous information to the decoder. By enforcing this constraint, input features are strictly represented as compositions of discrete symbols, ensuring that no continuous information is used for reconstruction.

The DB can be defined as a function $\mathbf{p}, \mathbf{v}_q = \mathrm{DB}(\mathbf{v})$, where:

- $\mathbf{p}$ represents a discrete distribution over tokens, facilitating **supervised training** with negative log-likelihood loss when labels are available for $z$.
- $\mathbf{v}_q$ is a quantized vector that serves as input to subsequent models or layers, such as the decoder in **unsupervised training** when labels are not provided.

Here, $\mathbf{p} \in [0,1]^{|V|}$ is a probability vector over the vocabulary $V$, with $\sum_{i=1}^{|V|} \mathbf{p}_i = 1$, and $|V|$, the vocabulary size, is treated as a hyperparameter. The discrete nature of the DB implies that the quantized vector belongs to a finite set $\mathbf{v}_q \in D$ , where $D = \{D[i]\}_{i=1}^{|V|}$ is a dictionary of embeddings.

This discrete computation introduces non-differentiability, which requires surrogate gradients to enable gradient-based optimization. By constraining the model output to a finite set of vectors, the model is forced to represent input features as compositions of dictionary vectors rather than a continuous representation Liu et al. (2022).

### 2.2.1 DISCRETE BOTTLENECK IMPLEMENTATIONS

The discrete bottleneck (DB) unifies methods like VQ-VAE, Gumbel-Softmax reparameterization, and Softmax/Argmax approximations as different implementations of the same underlying concept: quantization of latent variables. At their core, these methods estimate a probability over the latent variable $p_\theta(x, \tau)$, where $\tau$ is the temperature controlling the sharpness of the distribution.

If partial labels for the latent variable are available, the model can be trained using the negative log-likelihood loss. Otherwise, sampling from the distribution is necessary, which can be done via the Gumbel reparameterization trick with a sampling temperature $\tau_g$:

$$z = \text{one-hot}[\arg\max_i p_g(p_\theta(x, \tau), \tau_g)] \tag{6}$$

**Embedding-Based Quantization.** In **VQ-VAE**, the latent distribution is a one-hot vector, and the quantized vector is the nearest dictionary embedding to the input vector:

$$\mathbf{l}[i] = \|\mathbf{v} - D[i]\|, \ \mathbf{v}_q = D\left[\arg\min_i \mathbf{l}[i]\right], \ \mathbf{p}[i] = \begin{cases} 1 & \text{if } i = \arg\min_i \mathbf{l}[i] \\ 0 & \text{otherwise} \end{cases} \tag{7}$$

This is equivalent to sampling with both $\tau_g, \tau = 0$ in the equation above, i.e., a deterministic nearest-neighbor lookup. In the non-degenerate case, where $\tau > 0$, the softmax of the distance between the input vector and the dictionary embeddings is used, and the model can be trained with negative log-likelihood loss:

$$\mathbf{p}[i] = \frac{\exp(-\frac{\|\mathbf{v} - D[i]\|}{\tau})}{\sum_{j=1}^{|V|} \exp(-\frac{\|\mathbf{v} - D[j]\|}{\tau})}, \ \mathbf{v}_q = D\left[\arg\max_i \mathbf{p}[i]\right] \tag{8}$$

We study this as the **VQ-DB** in our experiments, an instance of embedding-based quantization where the probability vector $p$ is a function of the dictionary embeddings, $p = S(\cdot; D)$.

**Probability-Based Quantization.** In contrast to VQ-VAE, we study the Softmax/Argmax approximation, which corresponds to $\tau_g = 0$ and $\tau > 0$, but without dependence on dictionary embeddings. Additionally, we examine the Gumbel-Softmax discretizer, a non-degenerate case with both $\tau_g > 0$ and $\tau > 0$, introducing stochasticity while maintaining a differentiable approximation of the categorical variable.

The **Softmax DB** uses maximum likelihood decoding, where the quantized vector corresponds to the most likely token in the dictionary:

$$\mathbf{p}[i] = \frac{\exp(\frac{\mathbf{v}[i]}{\tau})}{\sum_{j=1}^{|V|} \exp(\frac{\mathbf{v}[j]}{\tau})}, \quad \mathbf{v}_q = D\left[\arg\max_i \mathbf{p}[i]\right] \tag{9}$$

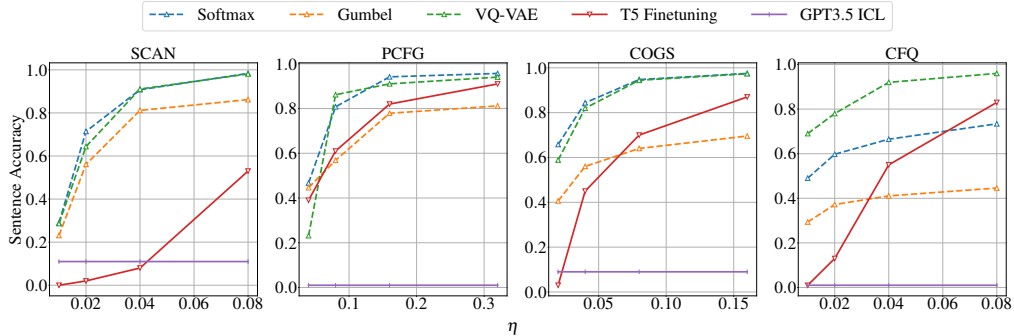

Figure 2: $Z$ Sentence Accuracy per supervision ratio ($\eta$). Dashed lines denote the best performance of $\Sigma$AE framework using different discretizers which shows consistent superiority of Softmax and VQ-VAE over the pretrained and in-context learning baselines.

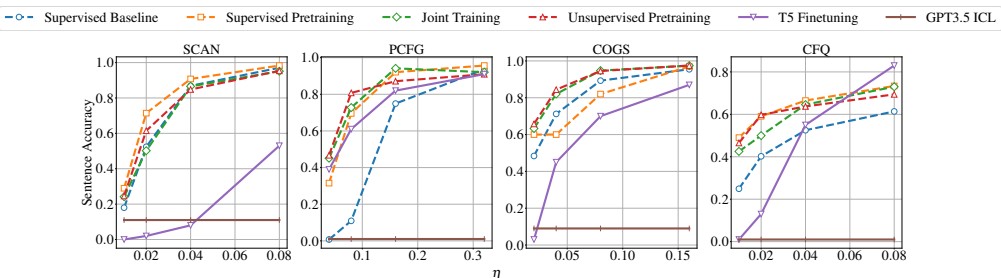

Figure 3: Results for Softmax Discrete Bottleneck – $Z$ Sentence Accuracy per supervision ratio ($\eta$). At least one training method with the $\Sigma$AE framework consistently outperforms the pretrained and in-context learning baselines except on the CFQ dataset at 8% supervision ratio. (ICL, flat line with fixed number (20) of in-context samples) and the supervised training baselines. Fine-tuned T5 on the CFQ dataset at an 8% supervision ratio outperforms our methods and all other baselines.

The **Gumbel DB** uses true categorical sampling for decoding:

$$\mathbf{p}[i] = \frac{\exp\left(\frac{1}{\tau_g}\left(\log(\frac{\exp(\frac{\mathbf{v}[i]}{\tau})}{\sum_{j=1}^{|V|}\exp(\frac{\mathbf{v}[j]}{\tau})}) + g_i\right)\right)}{\sum_{j=1}^{|V|}\exp\left(\frac{1}{\tau_g}\left(\log\frac{\exp(\frac{\mathbf{v}[j]}{\tau})}{\sum_{k=1}^{|V|}\exp(\frac{\mathbf{v}[k]}{\tau})} + g_j\right)\right)}, \quad \mathbf{v}_q = D\left[\arg\max_i \mathbf{p}[i]\right] \tag{10}$$

Here, $g_i$ is a sample from the Gumbel distribution, i.e., $g_i = -\log(-\log(u_i))$, where $u_i \sim$ Uniform$(0, 1)$, using the Gumbel reparameterization trick to translate the sampling into the argmax of noisy probabilities Jang et al. (2017).

These are examples of probability-based quantization, where the probability vector $\mathbf{p}$ is a function of the input vector $\mathbf{v}$, i.e., $\mathbf{p} = S(\mathbf{v})$, without dependence on dictionary embeddings.

## 2.3 TRAINING A SEQUENCE OF DISCRETE RANDOM VARIABLES

To learn a latent variable $\bar{Z}$, which represents a sequence of discrete tokens, we extend the single discrete random variable framework. The joint probability distribution over the sequence $\bar{Z} = (\bar{Z}_1, \cdots, \bar{Z}_T)$ given input $X = x$ is modeled as:

$$p_\theta(\bar{Z} = (\bar{Z}_1, \cdots \bar{Z}_T)|X = x) = \prod_{t=1}^{T} p_\theta(\bar{Z}_t|X = x, \bar{Z}_{<t}) \tag{11}$$

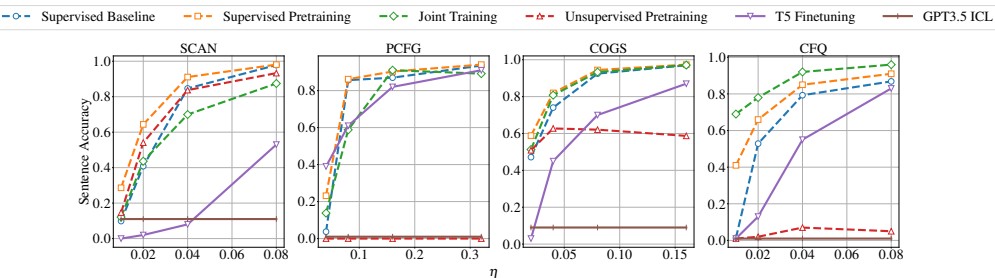

Figure 4: Results for VQ-VAE Discrete Bottleneck – $Z$ Sentence Accuracy per supervision ratio ($\eta$). Almost all training methods with the $\Sigma$AE framework consistently outperforms the pretrained and in-context learning baselines.

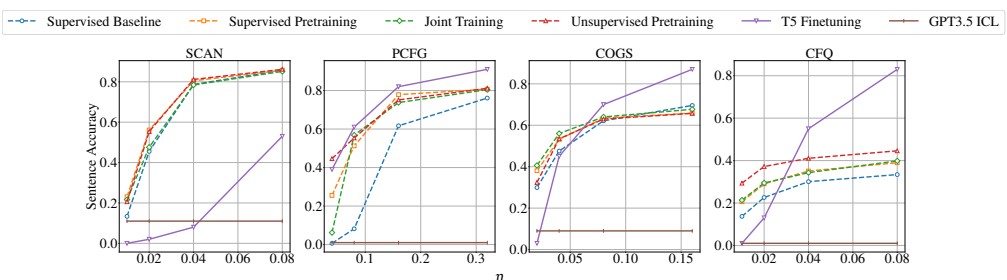

Figure 5: Results for Gumbel Discrete Bottleneck – $Z$ Sentence Accuracy per supervision ratio ($\eta$). Due to the inherent randomness of Gumbel DB, the prediction could be noisy, and the full sentence accuracy is more prone to fall as can be seen in the plots. However, you can find the token accuracy in the appendix A.7 showing a more stable measure of performance.

In symbolic systems, the sequence length $T$ is itself a random variable. More complex inputs are likely to be mapped to longer "sentences" in the emergent symbolic language.

We model this process by introducing a masking function $Z = M(\bar{Z})$, which effectively masks out the tokens that appear after the first integer representing the end of the sequence. Formally, the sequence $z_1, z_2, \cdots, z_T$ is defined as:

$$(z_1, z_2, \cdots, z_T) = (\bar{z}_1, \bar{z}_2, \cdots, \bar{z}_T) \odot \mathbf{m} \tag{12}$$

Here, $\mathbf{m} = \mathbf{m}(\bar{Z})$ is a binary mask that **depends on the sampled outputs**, where $\mathbf{m}[i] = 1$ if the EOS (End-of-Sequence) token has not been generated, and $\mathbf{m}[i] = 0$ otherwise. Substituting $Z = \bar{Z} \odot \mathbf{m}(\bar{Z})$ in the forward computation necessitates evaluating $\nabla_\theta l(Z) = \nabla_\theta l(\bar{Z} \odot \mathbf{m}(\bar{Z}))$, which requires differentiating the binary mask $\mathbf{m}$ during the backward pass. This challenge is addressed in the next section (2.3.1).

The choice of symbol for the EOS token is arbitrary but needs to be explicitly modeled. By penalizing the likelihood of the EOS token we can control the brevity of sentences generated in the emergent language.

### 2.3.1 HIDDEN SEQUENCE COLLAPSE IN SEQ2SEQ MODELS

In symbolic autoencoding, the encoder generates hidden tokens until an End-of-Sequence (EOS) token or a maximum length is reached. This process involves a discrete decision about when to halt generation, for which the model never receives gradient feedback. Specifically, in unsupervised training, the loss gradient doesn't directly inform the model that mistakenly assigning a high likelihood to EOS has a penalty beyond the negative log likelihood loss: it can prematurely stop the entire sequence generation.

In our early autoencoder trainings we empirically observed that the models tended to completely ignore the latent sequences, or rely excessively on the first tokens of the hidden representation,

leading to underutilization of subsequent tokens. This resulted in what we term **hidden sequence collapse**—a tendency for the model to terminate sequences prematurely and only train the decoder. This probem is similar in nature to the posterior collapse problem in VAEs for textual data, and has been reported in various forms in prior work Bowman et al. (2016); Havrylov & Titov (2020); Newman et al. (2020); Zhao et al. (2018). To address this, we developed a soft-masking mechanism that allows for gradient-based learning of when to halt generation.

### 2.3.2 EOS SOFT-MASKING – GRADIENT APPROXIMATION FOR HALTING THE GENERATION

In unsupervised training, sequences generated within a batch can have varying halting points. Typically, the generation continues until either the maximum sequence length is reached or the EOS token is produced. Tokens generated after the EOS are masked out using a binary mask $\mathbf{m}$ of length $T$ (the number of tokens) as defined in the previous section. The mask at time step $i$ can be rewritten recursively as a function of the mask at the previous step $i - 1$ and the current sampled output token, $\bar{z}_{i-1}$:

$$\mathbf{m}[i] = \begin{cases} 1 & \text{if } \bar{z}_{i-1} \neq \text{eos-token-id, and } \mathbf{m}[i-1] = 1 \\ 0 & \text{otherwise} \end{cases} \tag{13}$$

Applying the mask to the quantized vectors $\mathbf{v}_q^{<T}$ during the forward pass enforces a halting mechanism by setting vectors post-EOS to a fixed, padding embedding, $\mathbf{v}_q \leftarrow \mathbf{v}_q \odot \mathbf{m} + D[\texttt{<PAD>}] \odot (1 - \mathbf{m})$, thereby terminating the sequence generation. The challenge arises during the backward pass, as this mask is a non-differentiable output of the forward computation. To address this, we propose a gradient approximation for $\mathbf{m}$ that allows the model to learn the EOS effect through a feedback mechanism. To mitigate autoregressive collapse, we pass the gradients through $\mathbf{m}$ to $p_\theta(\bar{z}_k = \text{eos-token-id})$ as if $\mathbb{E}[\mathbf{m}[i]] = \prod_{k=1}^{i-1} (1 - p_\theta(\bar{z}_k = \text{eos-token-id}))$ had been the masking matrix in the forward computation. This approximation provides direct feedback on the EOS effect by simply assigning $\mathbf{m} \leftarrow \mathbf{m} + \mathbb{E}[\mathbf{m}] - \text{sg}(\mathbb{E}[\mathbf{m}])$. The derivation of this approximation is detailed in Appendix A.2.

## 3 EXPERIMENTS

### 3.1 TRAINING MODELS WITH AND WITHOUT SUPERVISION ON SEQ2SEQ TASKS

Given two symbolic systems $X$ and $Z$, we aim to learn the mappings $M_{xz}$ from $X$ to $Z$ and $M_{zx}$ from $Z$ to $X$. This is normally done with parallel data $D_{xz}$, where each input $x$ has a corresponding target $z$. $\Sigma$AE allows extracting information from unparallel data, by mapping data from the input space, to a latent space, and back to the input space for reconstruction.

We model the above system as an autoregressive decoder model, where the latent variable $Z$ is generated autoregressively from the input $X$. We incorporate a DB layer into each seq2seq model: $\text{DB}_x$ to $M_{zx}$ and $\text{DB}_z$ to $M_{xz}$, enabling both separate and joint training modes.

For parallel training data $(x, z) \in D_{xz}$ we do a **supervised training** step similar to common seq2seq training. Given input sequence $x$ and target sequence until step $t$, $z^{<t}$, the model $M_{xz}$ predicts a probability vector $\mathbf{p}_z^t$ for the $t$-th token $z^t$ and receives a loss (similarly for predicting the $x$ sequence):

$$\mathbf{p}_z^t, \mathbf{v}_z^t = \text{DB}_z(M_{xz}(x, z^{<t})), \ \mathcal{L}_{xz} = -\sum_t \log \mathbf{p}_z^t[z^t] \tag{14}$$

Given unlabeled data ($x \in D_x$ or $z \in D_z$), the models generate a latent sequence of quantized vectors ($\mathbf{v}_x^{<T_x} = \{\mathbf{v}_x^t\}_{t=0}^{T_x}$):

$$\mathbf{p}_x^t, \mathbf{p}_x^t = \text{DB}_x(M_{zx}(z, \mathbf{v}_x^{<t})) \tag{15}$$

These vectors are then used to reconstruct the original input:

$$\mathbf{p}_z^t, \mathbf{v}_z^t = \text{DB}_z(M_{xz}(\mathbf{v}_x^{<T_x}, z^{<t})), \tag{16}$$

using as the reconstruction loss $\mathcal{L}_{zxz} = -\sum_t \log \mathbf{p}_z^t[z^t]$. Similar steps are followed for the $x$ sequence. We call these **X Reconstruction** and **Z Reconstruction** modes, where we use unparallel data, $D_x$ or $D_z$, to minimize reconstruction losses $\mathcal{L}_{xzx}$ or $\mathcal{L}_{zxz}$.

To navigate this multi-objective optimization problem, we propose three scheduling strategies: **Joint Training** involves randomly selecting a batch from $D_{xz}$, $D_x$, or $D_z$ at each iteration and training in the corresponding mode. **Unsupervised Pretraining with Supervised Finetuning** starts with training on $D_x$ and $D_z$ until convergence, followed by fine-tuning on $D_{xz}$. Conversely, **Supervised Pretraining with Unsupervised Finetuning** trains on $D_{xz}$ until convergence, then shifts to fine-tuning on $D_x$ and $D_z$.

### 3.2 Experimental Setup

For our experiments, we utilized four seq2seq datasets: **SCAN** Lake & Baroni (2017), **PCFG SET** Hupkes et al. (2019), **CFQ** Keysers et al. (2019), and **COGS** Kim & Linzen (2020), chosen for their compositional complexity, controlled environments, and precise accuracy measures. We evaluated the framework on the aforementioned datasets, focusing on sentence accuracy (SA) and token accuracy (TA). Additional performance metrics are discussed in the appendix in Section A.5. More details on the datasets are provided in Section A.3.

### 3.3 Baselines

In our experiments, we compare the performance of the $\Sigma$AE framework against the following baselines: (1) **Supervised Fine-tuning of a Pretrained Model (T5 large)**, where a pretrained T5 model is fine-tuned on the available parallel data; (2) **In-context Learning (ICL) with a Large Language Model (GPT-3.5)**, which utilizes GPT-3.5 to perform tasks based on given context without explicit fine-tuning; and (3) **Supervised Training from Scratch**, where a model is trained from scratch on the available parallel data. Further details on the tasks, model architecture, and hyperparameters are provided in Section A.4

### 3.4 Experimental Results

To show the feasibility of symbolic autoencoding with straight-through gradients updates we performed an unsupervised autoencoding reconstruction experiment for each dataset and DB, and observed that the models successfully learned a compression of the input sequences, as shown in Table 1. The results are further detailed in Section A.8.1.

In the weakly supervised task, we simulated a Rosetta Stone-like scenario with a mix of parallel and unparallel data, varying the ratio of parallel data ($\eta$) to assess the framework's ability to balance and integrate supervised and unsupervised losses. Results for the Softmax, VQ-VAE, and Gumbel DB are detailed in Figures 3, 4, 5.

Figure 2 shows the maximum performance of $\Sigma$AE framework methods at each supervision ratio for different DBs. Our experiments demonstrated that the $\Sigma$AE framework can efficiently utilize small amounts of parallel data to improve performance on larger unparallel datasets. At each supervision ratio $\eta$, one of our scheduling methods from Section 3.1 consistently outperformed the supervised baselines. As expected, model accuracy improved with increased supervised data, narrowing the performance gap as accuracies converged to their maxima. An exception was observed in the CFQ dataset (for Softmax DB) at an 8% supervision ratio, where fine-tuning the T5-large model outperformed our methods. This is likely due to the CFQ dataset's closer resemblance to natural language question answering tasks, benefiting the T5 model, which is pretrained on similar tasks. Additional remarks on training dynamics and learning behavior are provided in Section A.6. A detailed analysis of the results and the full set of performance metrics, including other DBs, are presented in Section A.8.2.

Additionally, due to the inherent randomness of Gumbel DB, the prediction could be noisy, and the full sentence accuracy is more prone to fall as can be seen in Figure 5. However, results on token accuracy A.7 show a more stable measure of performance in which $\Sigma$AE framework methods outperform the baselines.

Table 1: Table of Test Autoregressive token accuracy ($Z$) (top) and Sentence Accuracy ($Z$) (bottom) on the unsupervised autoencoding task ($Z$ reconstruction). A high token accuracy is achieved across all datasets, showing the feasibility of learning discrete representations with gradient descent-based methods. Sentence accuracy is similar for Softmax and Gumbel DBs, while VQ DB shows makes errors on all but the SCAN dataset. SCAN dataset has the shortest average sentence length, which could explain the higher accuracy.

|  | SCAN | PCFG | COGS | CFQ |
|---|---|---|---|---|
| Softmax DB | 1.00 | 0.74 | 0.98 | 0.99 |
|  | 0.96 | 0.31 | 0.55 | 0.69 |
| Gumbel DB | 0.98 | 0.75 | 0.98 | 0.99 |
|  | 0.74 | 0.36 | 0.51 | 0.43 |
| VQ DB | 1.00 | 0.44 | 0.94 | 0.90 |
|  | 0.93 | 0.00 | 0.03 | 0.00 |

## 4 LIMITATIONS AND FUTURE WORK

Exploring autoencoding for sequences, especially in weakly supervised settings, reveals significant challenges and opportunities for advancement. While our approach effectively reduces unsupervised reconstruction loss, it does not always directly translate to improved model performance. One key limitation is the slower training pace of autoregressive models due to their sequential nature, which hinders parallelization. Moreover, our findings suggest unsupervised samples have less impact on accuracy compared to supervised ones, which need to be studied in future work. The $\Sigma$AE framework's adaptability to various seq2seq models and its applicability across different data modalities—from text to images, audio, or video—highlight its broad utility. This flexibility suggests numerous pathways for further exploration and application beyond the current study's scope.

## 5 CONCLUSION

In this study, we introduced a framework for learning latent variable models where the latent space is both discrete and sequential—representing sentences from an emergent symbolic system. We proposed a novel approach for training seq2seq models using non-parallel data. This was achieved by connecting two models through a discrete bottleneck, enabling the output sequence from one model to serve as the hidden representation for the other. This design creates a unique autoencoder architecture in which both the encoder and decoder are seq2seq models. To ensure the end-to-end trainability of this autoencoder within a gradient descent framework, we proposed gradient substitute and autoregressive masking techniques. Our unsupervised experiments validated the feasibility of training models within this discrete sequential autoencoder setup.

Further expanding on this concept we leveraged the symbolic autoencoders to train seq2seq models beyond parallel data and facilitate the use of non-parallel data. Demonstrating a practical application of our methodology, we presented evidence of performance improvements in weak supervision settings by utilizing unsupervised monolingual data, tested across four distinct seq2seq datasets.

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

## A APPENDIX

### A.1 REMARKS ON ΣAE FRAMEWORK

For all our straight-through gradient estimations, as training progresses, models become more confident in their predictions, resulting in more polarized score distributions. This polarization helps the models identify the most likely token with increasing certainty, making the scores sparser and improving the accuracy of gradient approximations.

While we only use symbolic autoencoding in reconstruction setups, the framework is adaptable to additional models and data sources. For instance, one could imagine models $M_{zy}, M_{yx}$, etc., each with their own supervised and reconstruction losses (e.g., $\mathcal{L}_{zy}, \mathcal{L}_{yxz}, \mathcal{L}_{xyx}$, etc.) to be optimized. Unlike some multi-task scenarios where individual tasks may appear independent or unrelated, in the ΣAE framework, improvement in one task can directly benefit others, creating a synergy that enhances overall performance.

### A.2 EOS GRADIENT APPROXIMATION

The EOS collapse phenomena can be explained by the model's lack of understanding of the EOS token's impact. Without explicit feedback, the model does not learn the importance of distributing information across the entire sequence. The EOS collapse phenomenon arises from the model's lack of understanding of the EOS token's role in sequence generation. Without explicit feedback, the model fails to learn the importance of distributing information evenly across the entire sequence. Instead, it observes that tokens appearing later in the sentence are randomly masked out, leading

to a tendency to concentrate all information into the first few tokens to ensure robust decoding. This behavior is loosely resembles to the effect of dropout Srivastava et al. (2014), where randomly masking information encourages models to learn more robust and compact representations.

In the $\Sigma$AE framework, we inform the model of the halting effect of the EOS token by approximating a gradient for the mask $\mathbf{m}$, which masks the tokens appearing after the first EOS token. This approximation is crucial for the model to learn the halting effect of the EOS token, essential for generating accurate sequences.

The $\mathbf{m}$ is 1 if the EOS token has not been generated and 0 otherwise:

$$\mathbf{m}[i] = \begin{cases} 1 & \text{if } \mathbf{m}[i-1] = 1 \text{ and } \bar{z}_k \neq \text{eos-token-id}) \\ 0 & \text{otherwise} \end{cases} \tag{17}$$

Hence, the binary random vector $\mathbf{m}$ is defined as:

$$p_\theta(\mathbf{m}[i] = 1) = (1 - p_\theta(\bar{z}_{i-1} = \text{eos-token-id}))\, p_\theta(\mathbf{m}[i-1] = 1)$$

$$= \prod_{k=1}^{i-1} (1 - p_\theta(\bar{z}_k = \text{eos-token-id})) . \tag{18}$$

Therefore the expected value of $\mathbf{m}$ is:

$$\mathbb{E}[\mathbf{m}[i]] = \prod_{k=1}^{i-1} (1 - p_\theta(\bar{z}_k = \text{eos-token-id})) \tag{19}$$

Our ablation studies showed that unsupervised training often failed due to hidden state collapse when this approximation was not used. Without this gradient approximation, the model struggled to learn effectively, highlighting the importance of this technique for successful training.

As training progresses, models become more confident in correctly predicting the EOS token, leading to more polarized probabilities. This makes the expected mask $\mathbb{E}[\mathbf{m}]$ a better approximate the true mask, thereby improving the accuracy of our approximation.

### A.3 DATASET DESCRIPTION AND EXAMPLES

We evaluated the $\Sigma$AE framework on four diverse datasets: SCAN, PCFG SET, CFQ, and COGS.

- SCAN Lake & Baroni (2017) is a simple language-driven navigation instruction task designed to evaluate the ability of neural models to learn compositional commands.
- PCFG SET Hupkes et al. (2019) is a synthetic dataset generated using probabilistic context-free grammars, aimed at testing the systematic generalization of models.
- CFQ Keysers et al. (2019) is a large-scale dataset of complex natural language questions and their corresponding SPARQL query against the Freebase knowledge base designed to measure the compositional generalization capabilities of semantic parsing models, with questions constructed to reflect the compositional structure of Freebase.
- COGS Kim & Linzen (2020): COGS is a dataset for evaluating the generalization of semantic parsing models to novel linguistic structures, emphasizing the model's ability to generalize from given sentences to new sentences that have similar syntactic structures but different lexical items or phrasal constructions.

Examples of samples from each dataset are provided in Table 2.

The selection of these datasets ensures a comprehensive and nuanced evaluation of the $\Sigma$AE framework. They facilitate direct evaluation of our approach, avoiding reliance on proxy metrics often used with larger datasets. Here, the mapping from $X$ to $Z$ is unique and non-reversible, with $Z$ typically being the longer sequence, serving as a reliable ground truth for $X$. Our study diverges from the typical use of these datasets for compositional generalization. Instead of focusing on out-of-distribution testing, we emphasize in-distribution performance assessment. We also conduct a bidirectional evaluation of both $M_{xz}$ and $M_{zx}$ models, reflecting realistic seq2seq model applications where translation in both directions holds equal significance, in line with the suggestions of Bastings et al. (2018).

Table 2: Example of Samples from Different Datasets, Including Average $X$ and $Z$ Lengths

| Dataset | Sample | Train set size | Parallel portion | Avg. $X$ Length | Avg. $Z$ Length |
|---------|--------|----------------|------------------|-----------------|-----------------|
| SCAN | **X:** look right thrice after run left
**Z:** I_TURN_LEFT I_RUN I_TURN_RIGHT I_LOOK I_TURN_RIGHT I_LOOK I_TURN_RIGHT I_LOOK | 13382 | 1% to 8% | 9.28 | 16.59 |
| PCFG SET | **X:** echo append append E18 C13 , L18 M17 , R1 L1 Y1 T18 J18
**Z:** E18 C13 L18 M17 R1 L1 Y1 T18 J18 J18 | 65734 | 4% to 32% | 39.05 | 24.69 |
| CFQ | **X:** Who influenced M1 's cinematographer , writer , and editor
**Z:** SELECT DISTINCT ?x0 WHERE
?x0 a ns:people.person.
?x0 ns:influence.influence_node.influenced ?x1.
?x1 ns:film.cinematographer.film M1.
?x1 ns:film.editor.film M1.
?x1 ns:film.writer.film M1. | 76594 | 2% to 16% | 15.70 | 71.21 |
| COGS | **X:** Olivia rolled Liam.
**Z:** roll . agent ( x_1 , Olivia ) AND roll . theme ( x_1 , Liam ) | 24155 | 1% to 8% | 16.04 | 55.40 |

These datasets were chosen for their controlled environments and precise accuracy measures, making them ideal for evaluating the framework's performance. Additionally, they are widely used for benchmarking symbolic modeling capabilities, as demonstrated in works such as Drozdov et al. (2023); Zhou et al. (2023), which highlight the complexity of these tasks for state-of-the-art models.

The autoregressive nature of our model means that a zxz-reconstruction pass on a trained model requires a number of forward passes proportional to the latent sequence length. For example, in SCAN, the computational cost of a zxz pass is approximately 10 times that of a supervised next-token prediction forward pass. This scaling reflects a fundamental computational trade-off inherent in autoregressive unsupervised sequence models. This limitation is not unique to $\Sigma$AE but is shared across autoregressive frameworks, including on-policy reinforcement learning, where rollouts are computationally expensive, and recurrent neural networks, whose computation scales with sequence length. While non-autoregressive approaches such as diffusion models could alleviate this scaling issue, they are not yet well-suited for discrete random variables and introduce other challenges.

This computational scaling also highlights the difficulty of studying longer sequences, such as those in natural language tasks. As sequence length and model size grow, the lack of supervised data (e.g., for teacher forcing) causes the compute and memory requirements to increase prohibitively with sequence length. Moreover, such tasks typically require domain-specific techniques and substantial engineering effort. Thus, this study focuses on providing a proof-of-concept under controlled conditions, laying the groundwork for future extensions to more complex, real-world applications.

**In-context learning (ICL) vs. Chain-of-thought (CoT) Prompting Baseline** It is helpful to compare the performance of our proposed method with ICL and CoT in state-of-the-art LLMs, however, it is not clear how CoT is applicable to the tasks considered in this work since explicit reasoning steps are not available for the datasets under study. Our synthetic datasets, structured as symbolic puzzles, lack the annotations required for CoT reasoning. Therefore, we opted for evaluating ICL by providing example pairs of $(X, Z)$ to assess to what extent the patterns could be learned in-context and extrapolated to the test sequences without the need for a neuro-symbolic auto-encoding. Our ICL baseline uses GPT3.5 which takes as input the concatenation of 20 pairs of $(X, Z)$, followed by the test sample's $X$ and is then prompted to generate the corresponding $Z$. We simply then compute the evaluation metrics from Sec. 3.2 using the generated $Z$ sequences, which constitute the results in Figures 2- 5.

## A.4 Details on Tasks, Model Architecture, and Hyperparameters

We conducted two sets of experiments on each dataset:

- **Unsupervised Training**: In this scenario, we only have access to unparallel data. The primary goal is to reconstruct $Z$ from a hidden discrete sequence. The framework matches the dictionary size and the maximum sequence length of the hidden representation to those of $X$. This setup evaluates the $\Sigma$AE framework's ability to compress the input sequence into a shorter sequence and accurately reconstruct it.
- **Weakly-supervised Training**: This scenario simulates the Rosetta Stone problem, where a small portion of the data is parallel, and the rest is unparallel. The objective is to leverage both parallel and unparallel data by minimizing unsupervised losses ($\mathcal{L}_{zxz}$ and $\mathcal{L}_{xzx}$)

and supervised losses ($\mathcal{L}_{zx}$ and $\mathcal{L}_{xz}$). We conduct experiments for each dataset and DB implementation, varying the supervision ratio $\eta = \frac{|D_{xz}|}{|D_{xz}|+|D_x|+|D_z|}$. This allows us to assess how effectively the framework uses limited parallel data to improve performance on larger unparallel datasets.

In our experiments with the $\Sigma$AE framework, we adopted a standardized model architecture and hyperparameter setting across all tasks to maintain consistency and focus on the framework's effectiveness. We utilized a six-layer transformer encoder–decoder model for $M_{xz}$ and $M_{zx}$, with 8 attention heads and a hidden size of 512. The model was trained using the Adam optimizer with learning rate reduction on loss plateau. We used greedy decoding consistently for all tasks, simplifying the decoding process and ensuring uniformity across experiments.

Model learning rates were manually chosen on the order of $10^{-3}$ or $10^{-4}$, to ensure a decrease in loss during the early stages of training. Hyperparameters were not extensively tuned. For each task, the same hyperparameters were used across different supervision ratios which are available in our configuration files in the code. This uniform approach underscores the framework's robustness, although we acknowledge that more nuanced tuning and regularization might yield higher performance.

In both unsupervised and supervised finetuning after pretraining approaches, a gradual curriculum shift is employed rather than an abrupt change. This involves slowly altering the probability distribution of the 'three-sided coin' used for batch selection in joint training, to transition smoothly from the initial training phase to the subsequent finetuning phase.

## A.5   EVALUATION METRICS

In assessing the performance of the $\Sigma$AE framework, we measured two distinct metrics: **sentence accuracy (SA)** and **token accuracy (TA)**. These metrics are designed to provide both a holistic and a detailed view of the model's capabilities. Sentence accuracy (SA) for a sample is counted as 1 if the entire sentence is correctly generated. Token accuracy (TA) is a more granular measure, where correctness of each predicted token in all sentences are measured separately. This metric allows for partial credit within a sentence, providing a finer understanding of the model's performance at the token level.

The token accuracy can be measured with two methods: We can *teacher-force* the correct previous tokens (as per the ground truth) to the model and measure its accuracy in predicting the next token. Alternatively, the model's previous outputs (which may or may not be correct) can be used as inputs for generating subsequent tokens. This *autoregressive* approach is generally more challenging than teacher-forcing.

Each $X$ has a unique corresponding $Z$, simplifying the assessment of accuracy in this direction, therefore, evaluating $M_{xz}$ performance is simply done by examining the *Autoregressive Z TA/SA*, directly measuring the model's capability to generate accurate $Z$ sequences. For a given $Z$, however, there could be multiple valid $X$ sequences. Therefore, to evaluate $M_{zx}$, we utilize the *Teacher-forced X TA*, which restricts the range of correct $X$ sequences for end tokens. Another approach is the *Reconstruction Z TA/SA*, where a model $M_{xz}$ maps a generated sequence $\hat{x}$ back to $Z$, and the accuracy of this reconstructed sequence serves as a proxy for the correctness of $\hat{x}$.

## A.6   REMARKS ON EXPERIMENTAL RESULTS

We note that the VQ DB faced a peculiar issue of numerical instability on the SCAN dataset after extended training periods (+500 epochs). This instability was addressed through weight clipping, suggesting that while $\Sigma$AE offers substantial benefits, optimizing stability and accuracy across different data representations and tasks may require tailored adjustments. These insights into performance variations across $X$ and $Z$ spaces not only highlight the framework's broad applicability but also pinpoint areas for future refinement to maximize the $\Sigma$AE framework's effectiveness.

The training time for a single epoch depends on the sequence length, dataset size, and the number of epochs, influenced by the autoregressive nature of our approach. Specifically, one autoencoding step requires a number of forward passes proportional to the latent sequence length of the autoregressive encoder. Consequently, the average training time per epoch scales with the product of: (latent

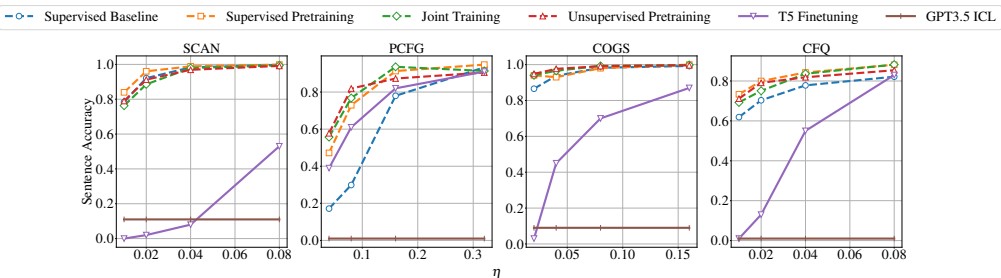

Figure 6: Results for Softmax Discrete Bottleneck – $Z$ Token Accuracy per supervision ratio ($\eta$).

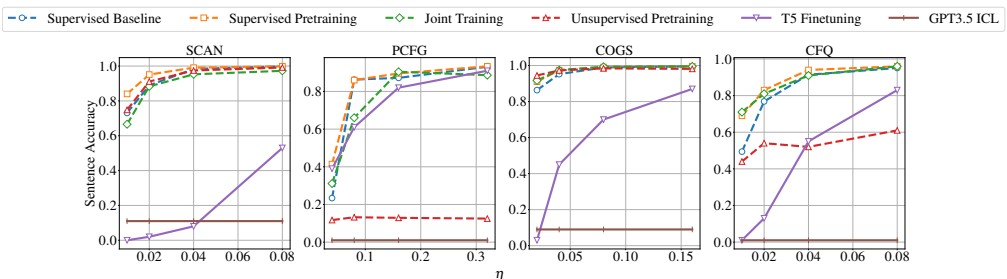

Figure 7: Results for VQ-VAE Discrete Bottleneck – $Z$ Token Accuracy per supervision ratio ($\eta$).

sequence length), (unsupervised-to-supervised sample size ratio), and (supervised training time per epoch).

### A.7 EXPERIMENT RESULTS

### A.8 WEAKLY SUPERVISED TRAINING - TOKEN ACCURACY FOR DIFFERENT DBS

Here we present the plots for different DBs using $\Sigma$AE framework methods in Figures **??**, **??**, **??** showing the token accuracy which is a more stable measure compared to full sentence accuracy.

#### A.8.1 UNSUPERVISED TRAINING RESULTS

In the unsupervised task, we trained the discrete autoencoder to compress and reconstruct $Z$ sequences without any supervised signal, evaluating the learnability of the discrete bottleneck using straight-through gradients. The results, summarized in Table 1, show that the Softmax DB achieved over 98% token accuracy on the SCAN, CFQ, and COGS datasets. Both the Gumbel and VQ DBs demonstrated similar effectiveness, indicating robustness in discrete autoencoding with straight-through gradients for sequence learning tasks. An exception to the high performance was the PCFG SET reconstruction task, where model performances were notably lower. This variation may be attributed to the unique symbolic nature of variables within the PCFG SET task, where basic tokenization assigns distinct representations to symbolically equivalent variables, leading to observed performance discrepancies.

#### A.8.2 WEAKLY SUPERVISED TRAINING RESULTS

In the $Z$ space, the Softmax DB consistently surpassed supervised baselines, significantly enhancing token and sentence accuracy across all datasets. For instance, with only 8% supervision on the PCFG SET dataset, token accuracy improved from below 15% to above 80%. While the Gumbel DB generally showed noisier training and slightly weaker performance, it still outperformed supervised baselines in most scenarios, except for a minor shortfall in the COGS dataset at a 16% supervision ratio. The VQ DB, despite showing a slight weaker performance in supervised baselines, improved the training similar to the Softmax and Gumbel DBs, achieving over 20% token accuracy on CFQ dataset at 2% supervision ratio.

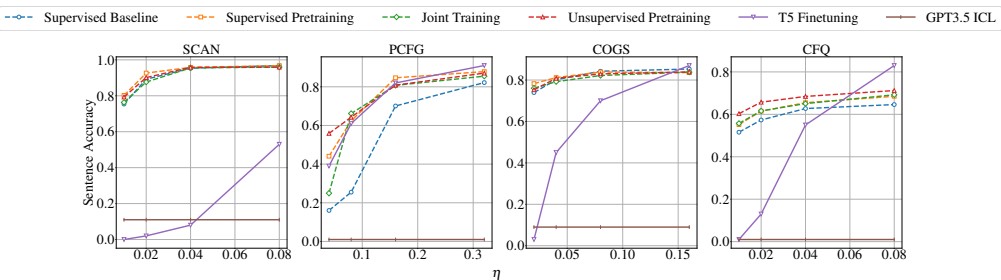

Figure 8: Results for Gumbel Discrete Bottleneck – $Z$ Token Accuracy per supervision ratio ($\eta$).

While no single Discrete Bottleneck or scheduling method universally outperforms others across all datasets and supervision ratios, for every dataset and $\eta$ value, at least one of our scheduling methods consistently surpasses the baseline performance. In other words, training within the $\Sigma$AE paradigm always enhances performance, though the optimal choice of the scheduling strategy depends on the task.

The $\Sigma$AE framework's impact extends into the $X$ space, where the Softmax, Gumbel, and VQ DBs exhibit performance boosts. Notably, the exception to this trend occurs with teacher-forced token accuracy in the SCAN dataset for the Softmax DB, indicating a unique challenge in this specific setting.

For all our experiments, we computed 95% confidence intervals via bootstrapped resampling of the test set, however they are too small to be visible on the plots. This performance analysis underscores the $\Sigma$AE framework's versatility and its capacity to leverage both unsupervised and weakly supervised data to enhance model training and performance across diverse seq2seq tasks.

We only measure the ICL and supervised finetuning of T5 baselines for Autoregressive Z TA and SA, as the teacher-forced X TA is not applicable to these baselines. The ICL baseline is a flat line with a fixed number of in-context samples (20) and the supervised finetuning of T5 is a single point at 100% supervision ratio.

We present the results of our experiments in the following tables. For Softmax discrete bottleneck, we present the results in the following tables:

- Table 3 Shows the performance of the Softmax DB on Autoregressive Z token accuracy, from test inputs
- Table 4 Shows the performance of the Softmax DB on Autoregressive Z sentence accuracy, from test inputs
- Table 5 shows the performance of the Softmax DB on the Autoregressive Z reconstruction token accuracy, after mapping to a hidden X
- Table 6 shows the performance of the Softmax DB on the Autoregressive Z reconstruction sentence accuracy, after mapping to a hidden X
- Table 7 shows the performance of the Softmax DB on the X token accuracy when teacher-forcing the previous inputs

For Gumbel discrete bottleneck, we present the results in the following tables:

- Table 8 Shows the performance of the Gumbel DB on Autoregressive Z token accuracy, from test inputs
- Table 9 Shows the performance of the Gumbel DB on Autoregressive Z sentence accuracy, from test inputs
- Table 10 shows the performance of the Gumbel DB on the Autoregressive Z reconstruction token accuracy, after mapping to a hidden X
- Table 11 shows the performance of the Gumbel DB on the Autoregressive Z reconstruction sentence accuracy, after mapping to a hidden X
- Table 12 shows the performance of the Gumbel DB on the X token accuracy when teacher-forcing the previous inputs

For VQ discrete bottleneck, we present the results in the following tables:

Table 3: Softmax DB – Autoregressive Z Token Accuracy. * These baselines are not concerned with the discretizer type and are not trained with our proposed discrete bottleneck. They will appear in all tables for comparison.

| SCAN | $\eta = 0.01$ | $\eta = 0.02$ | $\eta = 0.04$ | $\eta = 0.08$ | $\eta = 0.99$ |
|---|---|---|---|---|---|
| ICL (GPT3.5)* | – | – | – | – | 0.54 |
| T5 Finetuning* | 0.0 | 0.47 | 0.69 | 0.91 | – |
| Supervised Baseline | 0.78 | 0.92 | 0.98 | **1.00** | 1.00 |
| Joint training | 0.76 | 0.89 | 0.98 | 0.99 | —— |
| Supervised Pretraining | **0.84** | **0.96** | **0.99** | **1.00** | —— |
| Unsupervised Pretraining | 0.79 | 0.91 | 0.97 | 0.99 | —— |

| PCFG | $\eta = 0.04$ | $\eta = 0.08$ | $\eta = 0.16$ | $\eta = 0.32$ | $\eta = 0.99$ |
|---|---|---|---|---|---|
| ICL (GPT3.5) | – | – | – | – | 0.17 |
| T5 Finetuning | 0.50 | 0.74 | 0.85 | 0.93 | – |
| Supervised Baseline | 0.17 | 0.30 | 0.78 | 0.93 | 0.97 |
| Joint training | 0.56 | 0.77 | **0.94** | 0.91 | —— |
| Supervised Pretraining | 0.47 | 0.73 | 0.91 | **0.95** | —— |
| Unsupervised Pretraining | **0.58** | **0.82** | 0.87 | 0.91 | —— |

| COGS | $\eta = 0.02$ | $\eta = 0.04$ | $\eta = 0.08$ | $\eta = 0.16$ | $\eta = 0.99$ |
|---|---|---|---|---|---|
| ICL (GPT3.5) | – | – | – | – | 0.25 |
| T5 Finetuning | 0.35 | 0.72 | 0.95 | 0.99 | – |
| Supervised Baseline | 0.87 | 0.94 | 0.98 | 0.99 | 1.00 |
| Joint training | 0.94 | 0.97 | **0.99** | **1.00** | —— |
| Supervised Pretraining | 0.94 | 0.93 | 0.98 | **1.00** | —— |
| Unsupervised Pretraining | **0.95** | **0.98** | **0.99** | **1.00** | —— |

| CFQ | $\eta = 0.01$ | $\eta = 0.02$ | $\eta = 0.04$ | $\eta = 0.08$ | $\eta = 0.99$ |
|---|---|---|---|---|---|
| ICL (GPT3.5) | – | – | – | – | 0.26 |
| T5 Finetuning | 0.45 | 0.63 | **0.86** | **0.96** | – |
| Supervised Baseline | 0.62 | 0.70 | 0.78 | 0.82 | 0.86 |
| Joint training | 0.69 | 0.75 | 0.84 | 0.88 | —— |
| Supervised Pretraining | **0.73** | **0.80** | 0.84 | 0.88 | —— |
| Unsupervised Pretraining | 0.71 | 0.79 | 0.82 | 0.85 | —— |

- Table 13 Shows the performance of the VQ DB on Autoregressive Z token accuracy, from test inputs
- Table 14 Shows the performance of the VQ DB on Autoregressive Z sentence accuracy, from test inputs
- Table 15 shows the performance of the VQ DB on the Autoregressive Z reconstruction token accuracy, after mapping to a hidden X
- Table 16 shows the performance of the VQ DB on the Autoregressive Z reconstruction sentence accuracy, after mapping to a hidden X
- Table 17 shows the performance of the VQ DB on the X token accuracy when teacher-forcing the previous inputs

Table 4: Softmax DB – Autoregressive Z Sentence Accuracy

| SCAN | $\eta = 0.01$ | $\eta = 0.02$ | $\eta = 0.04$ | $\eta = 0.08$ | $\eta = 0.99$ |
|---|---|---|---|---|---|
| ICL (GPT3.5) | – | – | – | – | 0.11 |
| T5 Finetuning | 0.0 | 0.02 | 0.08 | 0.53 | – |
| Supervised Baseline | 0.18 | 0.52 | 0.87 | 0.97 | 1.00 |
| Joint training | 0.24 | 0.50 | 0.87 | 0.95 | —— |
| Supervised Pretraining | **0.29** | **0.71** | **0.91** | **0.98** | —— |
| Unsupervised Pretraining | 0.25 | 0.61 | 0.85 | 0.95 | —— |
| PCFG | $\eta = 0.04$ | $\eta = 0.08$ | $\eta = 0.16$ | $\eta = 0.32$ | $\eta = 0.99$ |
| ICL (GPT3.5) | – | – | – | – | 0.01 |
| T5 Finetuning | 0.39 | 0.61 | 0.82 | 0.91 | – |
| Supervised Baseline | 0.01 | 0.11 | 0.75 | 0.94 | 0.97 |
| Joint training | 0.45 | 0.73 | **0.94** | 0.92 | —— |
| Supervised Pretraining | 0.32 | 0.70 | 0.92 | **0.96** | —— |
| Unsupervised Pretraining | **0.47** | **0.81** | 0.87 | 0.91 | —— |
| COGS | $\eta = 0.02$ | $\eta = 0.04$ | $\eta = 0.08$ | $\eta = 0.16$ | $\eta = 0.99$ |
| ICL (GPT3.5) | – | – | – | – | 0.09 |
| T5 Finetuning | 0.03 | 0.45 | 0.70 | 0.87 | – |
| Supervised Baseline | 0.48 | 0.71 | 0.89 | 0.95 | 1.00 |
| Joint training | 0.63 | 0.82 | **0.95** | **0.97** | —— |
| Supervised Pretraining | 0.60 | 0.83 | 0.94 | **0.97** | —— |
| Unsupervised Pretraining | **0.66** | **0.84** | **0.95** | **0.97** | —— |
| CFQ | $\eta = 0.01$ | $\eta = 0.02$ | $\eta = 0.04$ | $\eta = 0.08$ | $\eta = 0.99$ |
| ICL (GPT3.5) | – | – | – | – | 0.01 |
| T5 Finetuning | 0.01 | 0.13 | 0.55 | 0.83 | – |
| Supervised Baseline | 0.25 | 0.40 | 0.53 | 0.61 | 0.69 |
| Joint training | 0.43 | 0.50 | 0.65 | **0.73** | —— |
| Supervised Pretraining | **0.49** | 0.59 | **0.66** | **0.73** | —— |
| Unsupervised Pretraining | 0.47 | **0.60** | 0.64 | 0.69 | —— |

Table 5: Softmax DB – Reconstruction Z TA

| SCAN | $\eta = 0.04$ | $\eta = 0.08$ | $\eta = 0.16$ | $\eta = 0.32$ | $\eta = 0.99$ |
|---|---|---|---|---|---|
| Supervised Baseline | 0.74 | 0.81 | 0.89 | 0.92 | 0.96 |
| Joint training | **0.99** | 0.98 | 0.98 | **0.97** | — |
| Supervised Pretraining | **0.99** | **0.99** | **0.99** | 0.97 | — |
| Unsupervised Pretraining | 0.98 | 0.83 | 0.93 | **0.97** | — |
| PCFG | $\eta = 0.04$ | $\eta = 0.08$ | $\eta = 0.16$ | $\eta = 0.32$ | $\eta = 0.99$ |
| Supervised Baseline | 0.37 | 0.50 | 0.74 | 0.78 | 0.83 |
| Joint training | 0.71 | **0.80** | 0.86 | **0.91** | — |
| Supervised Pretraining | 0.68 | 0.75 | 0.86 | 0.89 | — |
| Unsupervised Pretraining | **0.76** | 0.79 | **0.88** | 0.87 | — |
| COGS | $\eta = 0.02$ | $\eta = 0.04$ | $\eta = 0.08$ | $\eta = 0.16$ | $\eta = 0.99$ |
| Supervised Baseline | 0.92 | 0.95 | 0.97 | 0.98 | 0.99 |
| Joint training | 0.98 | **0.99** | **1.00** | **1.00** | — |
| Supervised Pretraining | 0.98 | 0.97 | 0.99 | **1.00** | — |
| Unsupervised Pretraining | **0.99** | **0.99** | **1.00** | **1.00** | — |
| CFQ | $\eta = 0.01$ | $\eta = 0.02$ | $\eta = 0.04$ | $\eta = 0.08$ | $\eta = 0.99$ |
| Supervised Baseline | 0.94 | 0.96 | 0.97 | 0.98 | 0.99 |
| Joint training | 0.97 | 0.97 | 0.98 | **0.99** | — |
| Supervised Pretraining | **0.98** | 0.98 | **0.99** | 0.99 | — |
| Unsupervised Pretraining | **0.98** | **0.99** | **0.99** | 0.99 | — |

Table 6: Softmax DB – Reconstruction Z SA

| SCAN | $\eta = 0.04$ | $\eta = 0.08$ | $\eta = 0.16$ | $\eta = 0.32$ | $\eta = 0.99$ |
|---|---|---|---|---|---|
| Supervised Baseline | 0.05 | 0.11 | 0.24 | 0.28 | 0.46 |
| Joint training | 0.82 | 0.76 | 0.72 | 0.65 | — |
| Supervised Pretraining | **0.90** | **0.81** | **0.91** | **0.66** | — |
| Unsupervised Pretraining | 0.84 | 0.18 | 0.41 | **0.66** | — |
| PCFG | $\eta = 0.04$ | $\eta = 0.08$ | $\eta = 0.16$ | $\eta = 0.32$ | $\eta = 0.99$ |
| Supervised Baseline | 0.01 | 0.09 | 0.20 | 0.28 | 0.35 |
| Joint training | **0.21** | **0.29** | **0.44** | **0.63** | — |
| Supervised Pretraining | 0.15 | 0.22 | 0.33 | 0.47 | — |
| Unsupervised Pretraining | 0.19 | 0.26 | **0.44** | 0.41 | — |
| COGS | $\eta = 0.02$ | $\eta = 0.04$ | $\eta = 0.08$ | $\eta = 0.16$ | $\eta = 0.99$ |
| Supervised Baseline | 0.02 | 0.04 | 0.35 | 0.48 | 0.57 |
| Joint training | 0.51 | 0.76 | **0.93** | **0.97** | — |
| Supervised Pretraining | 0.55 | 0.48 | 0.68 | 0.96 | — |
| Unsupervised Pretraining | **0.75** | **0.81** | 0.90 | 0.95 | — |
| CFQ | $\eta = 0.01$ | $\eta = 0.02$ | $\eta = 0.04$ | $\eta = 0.08$ | $\eta = 0.99$ |
| Supervised Baseline | 0.11 | 0.23 | 0.36 | 0.48 | 0.53 |
| Joint training | 0.29 | 0.36 | 0.50 | 0.61 | — |
| Supervised Pretraining | 0.36 | 0.44 | **0.54** | **0.62** | — |
| Unsupervised Pretraining | **0.40** | **0.51** | 0.52 | 0.60 | — |

Table 7: Softmax DB – Teacher-forced X TA

| SCAN | $\eta = 0.04$ | $\eta = 0.08$ | $\eta = 0.16$ | $\eta = 0.32$ | $\eta = 0.99$ |
|---|---|---|---|---|---|
| Supervised Baseline | **0.66** | **0.77** | **0.84** | 0.88 | **0.88** |
| Joint training | 0.57 | 0.66 | 0.78 | 0.84 | — |
| Supervised Pretraining | 0.39 | 0.58 | 0.70 | 0.82 | — |
| Unsupervised Pretraining | 0.50 | 0.45 | 0.69 | 0.81 | — |
| PCFG | $\eta = 0.04$ | $\eta = 0.08$ | $\eta = 0.16$ | $\eta = 0.32$ | $\eta = 0.99$ |
| Supervised Baseline | 0.41 | 0.50 | 0.53 | 0.57 | 0.65 |
| Joint training | **0.50** | **0.54** | 0.57 | 0.61 | — |
| Supervised Pretraining | 0.47 | 0.50 | 0.54 | 0.57 | — |
| Unsupervised Pretraining | 0.48 | 0.50 | **0.61** | **0.63** | — |
| COGS | $\eta = 0.02$ | $\eta = 0.04$ | $\eta = 0.08$ | $\eta = 0.16$ | $\eta = 0.99$ |
| Supervised Baseline | 0.87 | 0.95 | 0.98 | 0.99 | 1.00 |
| Joint training | **0.90** | **0.96** | 0.99 | 1.00 | — |
| Supervised Pretraining | 0.00 | 0.88 | 0.95 | **0.99** | — |
| Unsupervised Pretraining | 0.88 | **0.96** | 0.98 | 0.99 | — |
| CFQ | $\eta = 0.01$ | $\eta = 0.02$ | $\eta = 0.04$ | $\eta = 0.08$ | $\eta = 0.99$ |
| Supervised Baseline | 0.74 | 0.79 | 0.82 | 0.85 | 0.88 |
| Joint training | **0.77** | **0.80** | **0.83** | **0.85** | — |
| Supervised Pretraining | 0.73 | **0.80** | **0.83** | **0.85** | — |
| Unsupervised Pretraining | 0.71 | 0.78 | 0.82 | 0.84 | — |

Table 8: Gumbel DB – Autoregressive Z Token Accuracy

| SCAN | $\eta = 0.04$ | $\eta = 0.08$ | $\eta = 0.16$ | $\eta = 0.32$ | $\eta = 0.99$ |
|---|---|---|---|---|---|
| ICL (GPT3.5) | – | – | – | – | 0.54 |
| T5 Finetuning | 0.0 | 0.47 | 0.69 | 0.91 | – |
| Supervised Baseline | 0.75 | 0.89 | 0.95 | **0.97** | 0.97 |
| Joint training | 0.76 | 0.88 | 0.95 | 0.96 | —— |
| Supervised Pretraining | **0.80** | **0.93** | **0.96** | **0.97** | —— |
| Unsupervised Pretraining | 0.79 | 0.90 | **0.96** | 0.96 | —— |
| PCFG | $\eta = 0.04$ | $\eta = 0.08$ | $\eta = 0.16$ | $\eta = 0.32$ | $\eta = 0.99$ |
| ICL (GPT3.5) | – | – | – | – | 0.54 |
| T5 Finetuning | 0.0 | 0.47 | 0.69 | **0.91** | – |
| Supervised Baseline | 0.16 | 0.25 | 0.70 | 0.82 | 0.89 |
| Joint training | 0.25 | **0.66** | 0.81 | 0.86 | —— |
| Supervised Pretraining | 0.44 | 0.62 | **0.85** | 0.88 | —— |
| Unsupervised Pretraining | **0.56** | 0.64 | 0.81 | 0.87 | —— |
| COGS | $\eta = 0.02$ | $\eta = 0.04$ | $\eta = 0.08$ | $\eta = 0.16$ | $\eta = 0.99$ |
| ICL (GPT3.5) | – | – | – | – | 0.54 |
| T5 Finetuning | 0.0 | 0.47 | 0.69 | **0.91** | – |
| Supervised Baseline | 0.74 | 0.80 | **0.84** | 0.85 | 0.86 |
| Joint training | 0.76 | 0.79 | 0.82 | 0.84 | —— |
| Supervised Pretraining | **0.78** | **0.81** | **0.84** | 0.84 | —— |
| Unsupervised Pretraining | 0.75 | **0.81** | 0.83 | 0.84 | —— |
| CFQ | $\eta = 0.01$ | $\eta = 0.02$ | $\eta = 0.04$ | $\eta = 0.08$ | $\eta = 0.99$ |
| ICL (GPT3.5) | – | – | – | – | 0.54 |
| T5 Finetuning | 0.0 | 0.47 | **0.69** | **0.91** | – |
| Supervised Baseline | 0.52 | 0.57 | 0.63 | 0.65 | 0.65 |
| Joint training | 0.56 | 0.62 | 0.65 | 0.69 | —— |
| Supervised Pretraining | 0.55 | 0.61 | 0.65 | 0.68 | —— |
| Unsupervised Pretraining | **0.60** | **0.66** | 0.68 | 0.71 | —— |

Table 9: Gumbel DB – Autoregressive Z Sentence Accuracy

| SCAN | $\eta = 0.04$ | $\eta = 0.08$ | $\eta = 0.16$ | $\eta = 0.32$ | $\eta = 0.99$ |
|---|---|---|---|---|---|
| ICL (GPT3.5) | – | – | – | – | 0.11 |
| T5 Finetuning | 0.0 | 0.02 | 0.08 | 0.53 | – |
| Supervised Baseline | 0.13 | 0.46 | 0.79 | **0.86** | 0.89 |
| Joint training | 0.22 | 0.48 | 0.78 | 0.85 | —— |
| Supervised Pretraining | **0.23** | **0.56** | 0.80 | **0.86** | —— |
| Unsupervised Pretraining | 0.21 | 0.55 | **0.81** | **0.86** | —— |
| PCFG | $\eta = 0.04$ | $\eta = 0.08$ | $\eta = 0.16$ | $\eta = 0.32$ | $\eta = 0.99$ |
| ICL (GPT3.5) | – | – | – | – | 0.11 |
| T5 Finetuning | 0.0 | 0.02 | 0.08 | 0.53 | – |
| Supervised Baseline | 0.01 | 0.08 | 0.62 | 0.76 | 0.84 |
| Joint training | 0.06 | **0.57** | 0.74 | 0.80 | —— |
| Supervised Pretraining | 0.26 | 0.51 | **0.78** | **0.81** | —— |
| Unsupervised Pretraining | **0.45** | 0.55 | 0.75 | **0.81** | —— |
| COGS | $\eta = 0.02$ | $\eta = 0.04$ | $\eta = 0.08$ | $\eta = 0.16$ | $\eta = 0.99$ |
| ICL (GPT3.5) | – | – | – | – | 0.11 |
| T5 Finetuning | 0.0 | 0.02 | 0.08 | 0.53 | – |
| Supervised Baseline | 0.30 | 0.48 | 0.62 | **0.70** | 0.73 |
| Joint training | **0.41** | **0.56** | **0.64** | 0.68 | —— |
| Supervised Pretraining | 0.38 | 0.53 | 0.63 | 0.66 | —— |
| Unsupervised Pretraining | 0.32 | 0.53 | 0.63 | 0.66 | —— |
| CFQ | $\eta = 0.01$ | $\eta = 0.02$ | $\eta = 0.04$ | $\eta = 0.08$ | $\eta = 0.99$ |
| ICL (GPT3.5) | – | – | – | – | 0.11 |
| T5 Finetuning | 0.0 | 0.02 | 0.08 | **0.53** | – |
| Supervised Baseline | 0.14 | 0.23 | 0.30 | 0.33 | 0.34 |
| Joint training | 0.21 | 0.29 | 0.34 | 0.40 | —— |
| Supervised Pretraining | 0.21 | 0.29 | 0.35 | 0.39 | —— |
| Unsupervised Pretraining | **0.29** | **0.37** | 0.41 | **0.45** | —— |

Table 10: Gumbel DB – Reconstruction Z TA

| SCAN | $\eta = 0.04$ | $\eta = 0.08$ | $\eta = 0.16$ | $\eta = 0.32$ | $\eta = 0.99$ |
|---|---|---|---|---|---|
| Supervised Baseline | 0.74 | 0.78 | 0.86 | 0.90 | 0.94 |
| Joint training | 0.96 | 0.94 | 0.96 | 0.95 | — |
| Supervised Pretraining | 0.97 | 0.98 | 0.97 | 0.97 | — |
| Unsupervised Pretraining | 0.81 | 0.88 | 0.90 | 0.94 | — |
| PCFG | $\eta = 0.04$ | $\eta = 0.08$ | $\eta = 0.16$ | $\eta = 0.32$ | $\eta = 0.99$ |
| Supervised Baseline | 0.33 | 0.46 | 0.70 | 0.75 | 0.79 |
| Joint training | 0.32 | 0.58 | 0.73 | 0.83 | — |
| Supervised Pretraining | 0.56 | 0.63 | 0.72 | 0.81 | — |
| Unsupervised Pretraining | 0.57 | 0.75 | 0.82 | 0.85 | — |
| COGS | $\eta = 0.02$ | $\eta = 0.04$ | $\eta = 0.08$ | $\eta = 0.16$ | $\eta = 0.99$ |
| Supervised Baseline | 0.90 | 0.93 | 0.96 | 0.97 | 0.98 |
| Joint training | 0.96 | 0.98 | 0.98 | 0.99 | — |
| Supervised Pretraining | 0.96 | 0.97 | 0.98 | 0.99 | — |
| Unsupervised Pretraining | 0.97 | 0.98 | 0.99 | 0.99 | — |
| CFQ | $\eta = 0.01$ | $\eta = 0.02$ | $\eta = 0.04$ | $\eta = 0.08$ | $\eta = 0.99$ |
| Supervised Baseline | 0.92 | 0.94 | 0.95 | 0.96 | 0.97 |
| Joint training | 0.95 | 0.96 | 0.96 | 0.97 | — |
| Supervised Pretraining | 0.94 | 0.95 | 0.96 | 0.97 | — |
| Unsupervised Pretraining | 0.98 | 0.98 | 0.98 | 0.98 | — |

Table 11: Gumbel DB – Reconstruction Z SA

| SCAN | $\eta = 0.04$ | $\eta = 0.08$ | $\eta = 0.16$ | $\eta = 0.32$ | $\eta = 0.99$ |
|---|---|---|---|---|---|
| Supervised Baseline | 0.05 | 0.08 | 0.17 | 0.27 | 0.30 |
| Joint training | 0.60 | 0.43 | 0.54 | 0.54 | — |
| Supervised Pretraining | 0.65 | 0.71 | 0.68 | 0.62 | — |
| Unsupervised Pretraining | 0.11 | 0.18 | 0.39 | 0.55 | — |
| PCFG | $\eta = 0.04$ | $\eta = 0.08$ | $\eta = 0.16$ | $\eta = 0.32$ | $\eta = 0.99$ |
| Supervised Baseline | 0.01 | 0.06 | 0.20 | 0.25 | 0.21 |
| Joint training | 0.01 | 0.11 | 0.19 | 0.38 | — |
| Supervised Pretraining | 0.07 | 0.11 | 0.15 | 0.25 | — |
| Unsupervised Pretraining | 0.05 | 0.21 | 0.32 | 0.42 | — |
| COGS | $\eta = 0.02$ | $\eta = 0.04$ | $\eta = 0.08$ | $\eta = 0.16$ | $\eta = 0.99$ |
| Supervised Baseline | 0.02 | 0.02 | 0.20 | 0.27 | 0.34 |
| Joint training | 0.29 | 0.48 | 0.59 | 0.61 | — |
| Supervised Pretraining | 0.24 | 0.42 | 0.56 | 0.60 | — |
| Unsupervised Pretraining | 0.30 | 0.46 | 0.59 | 0.63 | — |
| CFQ | $\eta = 0.01$ | $\eta = 0.02$ | $\eta = 0.04$ | $\eta = 0.08$ | $\eta = 0.99$ |
| Supervised Baseline | 0.05 | 0.10 | 0.17 | 0.21 | 0.21 |
| Joint training | 0.10 | 0.14 | 0.21 | 0.25 | — |
| Supervised Pretraining | 0.07 | 0.13 | 0.19 | 0.27 | — |
| Unsupervised Pretraining | 0.26 | 0.27 | 0.31 | 0.34 | — |

Table 12: Gumbel DB – Teacher-forced X TA

| SCAN | $\eta = 0.04$ | $\eta = 0.08$ | $\eta = 0.16$ | $\eta = 0.32$ | $\eta = 0.99$ |
|---|---|---|---|---|---|
| Supervised Baseline | 0.66 | 0.76 | 0.84 | 0.87 | 0.88 |
| Joint training | 0.60 | 0.70 | 0.78 | 0.85 | — |
| Supervised Pretraining | 0.40 | 0.63 | 0.76 | 0.84 | — |
| Unsupervised Pretraining | 0.36 | 0.64 | 0.62 | 0.76 | — |
| PCFG | $\eta = 0.04$ | $\eta = 0.08$ | $\eta = 0.16$ | $\eta = 0.32$ | $\eta = 0.99$ |
| Supervised Baseline | 0.39 | 0.48 | 0.51 | 0.55 | 0.57 |
| Joint training | 0.38 | 0.49 | 0.55 | 0.58 | — |
| Supervised Pretraining | 0.45 | 0.50 | 0.52 | 0.56 | — |
| Unsupervised Pretraining | 0.43 | 0.53 | 0.55 | 0.57 | — |
| COGS | $\eta = 0.02$ | $\eta = 0.04$ | $\eta = 0.08$ | $\eta = 0.16$ | $\eta = 0.99$ |
| Supervised Baseline | 0.84 | 0.93 | 0.97 | 0.98 | 0.99 |
| Joint training | 0.88 | 0.95 | 0.98 | 0.99 | — |
| Supervised Pretraining | 0.86 | 0.93 | 0.97 | 0.98 | — |
| Unsupervised Pretraining | 0.85 | 0.94 | 0.98 | 0.99 | — |
| CFQ | $\eta = 0.01$ | $\eta = 0.02$ | $\eta = 0.04$ | $\eta = 0.08$ | $\eta = 0.99$ |
| Supervised Baseline | 0.71 | 0.77 | 0.80 | 0.83 | 0.85 |
| Joint training | 0.75 | 0.79 | 0.81 | 0.84 | — |
| Supervised Pretraining | 0.72 | 0.78 | 0.81 | 0.84 | — |
| Unsupervised Pretraining | 0.69 | 0.77 | 0.81 | 0.84 | — |

Table 13: VQ DB – Autoregressive Z Token Accuracy

| SCAN | $\eta = 0.04$ | $\eta = 0.08$ | $\eta = 0.16$ | $\eta = 0.32$ | $\eta = 0.99$ |
|---|---|---|---|---|---|
| ICL (GPT3.5) | – | – | – | – | 0.54 |
| T5 Finetuning | 0.0 | 0.47 | 0.69 | 0.91 | – |
| Supervised Baseline | 0.73 | 0.89 | 0.98 | 1.00 | 1.00 |
| Joint training | 0.67 | 0.88 | 0.95 | 0.97 | —— |
| Supervised Pretraining | 0.84 | 0.95 | 0.99 | 1.00 | —— |
| Unsupervised Pretraining | 0.75 | 0.91 | 0.97 | 0.99 | —— |
| **PCFG** | $\eta = 0.04$ | $\eta = 0.08$ | $\eta = 0.16$ | $\eta = 0.32$ | $\eta = 0.99$ |
| ICL (GPT3.5) | – | – | – | – | 0.54 |
| T5 Finetuning | 0.0 | 0.47 | 0.69 | 0.91 | – |
| Supervised Baseline | 0.23 | 0.86 | 0.87 | 0.93 | 0.93 |
| Joint training | 0.31 | 0.66 | 0.90 | 0.89 | —— |
| Supervised Pretraining | 0.41 | 0.86 | 0.90 | 0.93 | —— |
| Unsupervised Pretraining | 0.12 | 0.13 | 0.13 | 0.13 | —— |
| **COGS** | $\eta = 0.02$ | $\eta = 0.04$ | $\eta = 0.08$ | $\eta = 0.16$ | $\eta = 0.99$ |
| ICL (GPT3.5) | – | – | – | – | 0.54 |
| T5 Finetuning | 0.0 | 0.47 | 0.69 | 0.91 | – |
| Supervised Baseline | 0.86 | 0.95 | 0.99 | 1.00 | 1.00 |
| Joint training | 0.92 | 0.97 | 0.99 | 0.99 | —— |
| Supervised Pretraining | 0.91 | 0.97 | 0.99 | 1.00 | —— |
| Unsupervised Pretraining | 0.94 | 0.97 | 0.98 | 0.98 | —— |
| **CFQ** | $\eta = 0.01$ | $\eta = 0.02$ | $\eta = 0.04$ | $\eta = 0.08$ | $\eta = 0.99$ |
| ICL (GPT3.5) | – | – | – | – | 0.54 |
| T5 Finetuning | 0.0 | 0.47 | 0.69 | 0.91 | – |
| Supervised Baseline | 0.49 | 0.77 | 0.91 | 0.95 | 0.84 |
| Joint training | 0.71 | 0.81 | 0.91 | 0.96 | —— |
| Supervised Pretraining | 0.69 | 0.83 | 0.94 | 0.96 | —— |
| Unsupervised Pretraining | 0.44 | 0.54 | 0.52 | 0.61 | —— |

Table 14: VQ DB – Autoregressive Z Sentence Accuracy

| SCAN | $\eta = 0.04$ | $\eta = 0.08$ | $\eta = 0.16$ | $\eta = 0.32$ | $\eta = 0.99$ |
|---|---|---|---|---|---|
| ICL (GPT3.5) | – | – | – | – | 0.11 |
| T5 Finetuning | 0.0 | 0.02 | 0.08 | 0.53 | – |
| Supervised Baseline | 0.10 | 0.41 | 0.85 | 0.98 | 1.00 |
| Joint training | 0.12 | 0.44 | 0.70 | 0.87 | —— |
| Supervised Pretraining | 0.29 | 0.64 | 0.91 | 0.98 | —— |
| Unsupervised Pretraining | 0.15 | 0.54 | 0.84 | 0.93 | —— |
| PCFG | $\eta = 0.04$ | $\eta = 0.08$ | $\eta = 0.16$ | $\eta = 0.32$ | $\eta = 0.99$ |
| ICL (GPT3.5) | – | – | – | – | 0.11 |
| T5 Finetuning | 0.0 | 0.02 | 0.08 | 0.53 | – |
| Supervised Baseline | 0.04 | 0.86 | 0.87 | 0.94 | 0.91 |
| Joint training | 0.14 | 0.59 | 0.91 | 0.89 | —— |
| Supervised Pretraining | 0.23 | 0.86 | 0.90 | 0.94 | —— |
| Unsupervised Pretraining | 0.00 | 0.00 | 0.00 | 0.00 | —— |
| COGS | $\eta = 0.02$ | $\eta = 0.04$ | $\eta = 0.08$ | $\eta = 0.16$ | $\eta = 0.99$ |
| ICL (GPT3.5) | – | – | – | – | 0.11 |
| T5 Finetuning | 0.0 | 0.02 | 0.08 | 0.53 | – |
| Supervised Baseline | 0.47 | 0.74 | 0.92 | 0.97 | 0.84 |
| Joint training | 0.51 | 0.81 | 0.93 | 0.97 | —— |
| Supervised Pretraining | 0.59 | 0.82 | 0.94 | 0.97 | —— |
| Unsupervised Pretraining | 0.51 | 0.63 | 0.62 | 0.59 | —— |
| CFQ | $\eta = 0.01$ | $\eta = 0.02$ | $\eta = 0.04$ | $\eta = 0.08$ | $\eta = 0.99$ |
| ICL (GPT3.5) | – | – | – | – | 0.11 |
| T5 Finetuning | 0.0 | 0.02 | 0.08 | 0.53 | – |
| Supervised Baseline | 0.01 | 0.53 | 0.79 | 0.87 | 0.25 |
| Joint training | 0.69 | 0.78 | 0.92 | 0.96 | —— |
| Supervised Pretraining | 0.41 | 0.66 | 0.85 | 0.91 | —— |
| Unsupervised Pretraining | 0.01 | 0.02 | 0.07 | 0.05 | —— |

Table 15: VQ DB – Reconstruction Z TA

| SCAN | $\eta = 0.04$ | $\eta = 0.08$ | $\eta = 0.16$ | $\eta = 0.32$ | $\eta = 0.99$ |
|---|---|---|---|---|---|
| Supervised Baseline | 0.73 | 0.77 | 0.83 | 0.91 | 0.99 |
| Joint training | 0.95 | 0.95 | 0.95 | 0.96 | — |
| Supervised Pretraining | 0.99 | 0.98 | 0.98 | 0.97 | — |
| Unsupervised Pretraining | 0.86 | 0.97 | 0.99 | 0.99 | — |
| PCFG | $\eta = 0.04$ | $\eta = 0.08$ | $\eta = 0.16$ | $\eta = 0.32$ | $\eta = 0.99$ |
| Supervised Baseline | 0.35 | 0.67 | 0.77 | 0.81 | 0.53 |
| Joint training | 0.59 | 0.77 | 0.91 | 0.93 | — |
| Supervised Pretraining | 0.68 | 0.81 | 0.83 | 0.89 | — |
| Unsupervised Pretraining | 0.32 | 0.38 | 0.32 | 0.32 | — |
| COGS | $\eta = 0.02$ | $\eta = 0.04$ | $\eta = 0.08$ | $\eta = 0.16$ | $\eta = 0.99$ |
| Supervised Baseline | 0.93 | 0.98 | 0.99 | 1.00 | 0.98 |
| Joint training | 0.97 | 0.99 | 0.99 | 1.00 | — |
| Supervised Pretraining | 0.97 | 0.98 | 0.99 | 1.00 | — |
| Unsupervised Pretraining | 0.98 | 0.99 | 0.98 | 0.98 | — |
| CFQ | $\eta = 0.01$ | $\eta = 0.02$ | $\eta = 0.04$ | $\eta = 0.08$ | $\eta = 0.99$ |
| Supervised Baseline | 0.89 | 0.95 | 0.98 | 0.98 | 0.98 |
| Joint training | 0.97 | 0.98 | 0.99 | 0.99 | — |
| Supervised Pretraining | 0.97 | 0.98 | 0.99 | 0.99 | — |
| Unsupervised Pretraining | 0.94 | 0.94 | 0.94 | 0.94 | — |

Table 16: VQ DB – Reconstruction Z SA

| SCAN | $\eta = 0.04$ | $\eta = 0.08$ | $\eta = 0.16$ | $\eta = 0.32$ | $\eta = 0.99$ |
|---|---|---|---|---|---|
| Supervised Baseline | 0.05 | 0.12 | 0.29 | 0.61 | 0.91 |
| Joint training | 0.40 | 0.38 | 0.39 | 0.52 | — |
| Supervised Pretraining | 0.89 | 0.76 | 0.71 | 0.66 | — |
| Unsupervised Pretraining | 0.17 | 0.67 | 0.90 | 0.85 | — |
| PCFG | $\eta = 0.04$ | $\eta = 0.08$ | $\eta = 0.16$ | $\eta = 0.32$ | $\eta = 0.99$ |
| Supervised Baseline | 0.02 | 0.18 | 0.19 | 0.25 | 0.12 |
| Joint training | 0.10 | 0.30 | 0.64 | 0.75 | — |
| Supervised Pretraining | 0.13 | 0.33 | 0.39 | 0.56 | — |
| Unsupervised Pretraining | 0.00 | 0.00 | 0.00 | 0.00 | — |
| COGS | $\eta = 0.02$ | $\eta = 0.04$ | $\eta = 0.08$ | $\eta = 0.16$ | $\eta = 0.99$ |
| Supervised Baseline | 0.32 | 0.68 | 0.78 | 0.95 | 0.39 |
| Joint training | 0.41 | 0.68 | 0.86 | 0.93 | — |
| Supervised Pretraining | 0.35 | 0.69 | 0.88 | 0.96 | — |
| Unsupervised Pretraining | 0.44 | 0.57 | 0.52 | 0.53 | — |
| CFQ | $\eta = 0.01$ | $\eta = 0.02$ | $\eta = 0.04$ | $\eta = 0.08$ | $\eta = 0.99$ |
| Supervised Baseline | 0.01 | 0.19 | 0.51 | 0.52 | 0.22 |
| Joint training | 0.27 | 0.40 | 0.57 | 0.67 | — |
| Supervised Pretraining | 0.19 | 0.41 | 0.56 | 0.71 | — |
| Unsupervised Pretraining | 0.00 | 0.01 | 0.00 | 0.00 | — |

Table 17: VQ DB – Teacher-forced X TA

| SCAN | $\eta = 0.04$ | $\eta = 0.08$ | $\eta = 0.16$ | $\eta = 0.32$ | $\eta = 0.99$ |
|---|---|---|---|---|---|
| Supervised Baseline | 0.65 | 0.73 | 0.81 | 0.86 | 0.88 |
| Joint training | 0.56 | 0.67 | 0.74 | 0.84 | — |
| Supervised Pretraining | 0.41 | 0.62 | 0.71 | 0.78 | — |
| Unsupervised Pretraining | 0.27 | 0.61 | 0.57 | 0.79 | — |
| PCFG | $\eta = 0.04$ | $\eta = 0.08$ | $\eta = 0.16$ | $\eta = 0.32$ | $\eta = 0.99$ |
| Supervised Baseline | 0.32 | 0.52 | 0.55 | 0.58 | 0.62 |
| Joint training | 0.43 | 0.54 | 0.59 | 0.65 | — |
| Supervised Pretraining | 0.48 | 0.53 | 0.55 | 0.58 | — |
| Unsupervised Pretraining | 0.33 | 0.34 | 0.33 | 0.33 | — |
| COGS | $\eta = 0.02$ | $\eta = 0.04$ | $\eta = 0.08$ | $\eta = 0.16$ | $\eta = 0.99$ |
| Supervised Baseline | 0.85 | 0.96 | 0.99 | 0.99 | 0.97 |
| Joint training | 0.88 | 0.95 | 0.98 | 0.99 | — |
| Supervised Pretraining | 0.85 | 0.93 | 0.98 | 0.99 | — |
| Unsupervised Pretraining | 0.88 | 0.93 | 0.93 | 0.93 | — |
| CFQ | $\eta = 0.01$ | $\eta = 0.02$ | $\eta = 0.04$ | $\eta = 0.08$ | $\eta = 0.99$ |
| Supervised Baseline | 0.60 | 0.77 | 0.83 | 0.86 | 0.88 |
| Joint training | 0.74 | 0.78 | 0.83 | 0.86 | — |
| Supervised Pretraining | 0.68 | 0.78 | 0.84 | 0.86 | — |
| Unsupervised Pretraining | 0.49 | 0.55 | 0.53 | 0.53 | — |

