# OpenReview forum: "Symbolic Autoencoding with Straight-Through Gradient Approximations"
_ICLR.cc/2025/Conference — Submitted to ICLR 2025_

### Official Review · Reviewer_H3MJ · 2024-10-31

**Soundness:** 2
**Presentation:** 1
**Contribution:** 2
**Rating:** 3
**Confidence:** 3

**Summary:**

This work addresses the general problem of translating between two symbolic systems, or, as the authors put it, the problem of text sequence transduction, when little parallel data is available. To tackle it, the authors consider an autoencoder framework, in which each symbolic system corresponds to the latent space of the other. In other words, the authors map sequences of tokens from one of the symbolic systems into a latent space whose elements are also defined to be token sequences, albeit (identified to be) from the second symbolic system. Key here is that this (unsupervised) mapping is done in both directions during training and only requires non-parallel data. Given this setting, the authors leverage the scarce parallel data to align the latent representations of each symbolic system.

The authors test their approach in four text sequence transduction tasks and demonstrate improvements over the baselines in scarce parallel data limits.

**Strengths:**

This work tackles a key problem inherent to many machine translation tasks, namely that state-of-the-art models require large high-quality parallel corpora, which in many applications are simply missing.

The proposed approach is interesting and (to my knowledge) original, in the sense that

- I had not read about symbolic system pairs treated as latent spaces of each other; and
- as the authors point out, it corresponds to the natural end-to-end extension of back-translation.

I also found the soft-masking approximation used to learn the halting mechanism of the hidden representation interesting.

**Weaknesses:**

Despite its merits, this work has a number of weaknesses, the most important of which have to do with the experimental section. Let me start by highlighting these.

- *Unconvincing results*: even within the setup proposed by the authors, the results are unconvincing. Indeed, as can be seen in Figure 4, the supervised baseline performs on par with the proposed method in *two out of four datasets, at already the second smallest supervision ratio*. It performs on par with the proposed method in three of the datasets, at the third supervision ratio. Or am I missing something?

Given that no information is provided about the training times of the proposed method, specially as compared with those of the supervised baseline, which is trained on a significantly smaller dataset, it is not clear what the advantage is in leveraging the non-parallel data to begin with.

- *Insufficient baselines*: in the related work section the authors highlight the connection of the proposed method with back-translation, which is depicted as the simplest approach which can leverage parallel and non-parallel data, specially in the general text sequence transduction setting studied by the authors. However, the latter is not considered as a baseline. Therefore, *the authors do not compare with any method that makes use of both parallel and non-parallel data*. This lack of comparison makes difficult the assesment of the proposed methology.

Similarly, there is little to no detail regarding the ICL baseline, whose performance is not discussed in the paper. It is known, for example, that ICL strongly depends on both the number and quality of examples (see e.g. Agrawal et al. [2023], Vilar et al. [2023]). I also have the feeling that the NLP community has moved from ICL to Chain-of-thought prompting, which makes the latter a more reasonable baseline. Note that the latter has been used in semantic parsing tasks (Tai et al [2023]).

- *Unconventional tasks*: Natural language translation and semantic parsing in low-resource settings are two major instances of the problem that could in principle be tackled by the proposed methodology. Nevertheless, the authors do not consider any of these.

In sum, these weaknesses have as a consequence that the reader does not fully grasp the relevance and usefulness of the proposed methodology.

Other weaknesses:

- *The mathematical notation is very confusing* and could significantly be improved. Some examples follow:

1. unless I am misunderstanding sth, the notation for probability distribution and density is constantly being exchanged (see e.g. 117 vs 121);
2. The notation for probability of an event changes within the same equation (see eq.18) and doesn’t match that in the main text;
3. Similar to the point above, see the eq. in lines 360 and 361. Instead of $\mathbb{P}$ the authors could use sth like $p_\theta(z=<EOS>|x)$ and thus make clear how the soft-mask approximation works. I might however be misunderstanding what the authors meant and, if so, I apologize;
4. again, unless I am misunderstanding, the class probabilities of the categorical distribution are labelled with both $\mathbf{p}$ and $\mathbf{s}$ in the same equations (see e.g. Eqs 8, 9 and 10).

- *The organisation of the paper could also be improved*: indeed, the authors spend a lot of time discussing how to optimise models which sample discrete variables. This procedure, whose description takes about 3 pages, is, in my view, well known by the ML community by now. Therefore, its discussion could probably be moved into the Appendix. In contrast, the reader would benefit from having e.g. the description of the dataset back into the main text.

- *Lack of references*: The authors make emphasis on the novelty of their autoencoder framework for latent sequences of symbols, but do not make reference to several other such methods as e.g. those of Zhao et al. [2018], Kaiser et al. [2018] or Sanchez et al. [2023].

**References**:

- Agrawal et al. [2023]: In-context Examples Selection for Machine Translation
- Vilar et al. [2023]: Prompting PaLM for Translation: Assessing Strategies and Performance
- Tai et al. [2023]: Exploring Chain of Thought Style Prompting for Text-to-SQL
- Ko et al. [2021]: Adapting High-resource NMT Models to Translate Low-resource Related
- Zhao et al. [2018]: Unsupervised Discrete Sentence Representation Learning for Interpretable Neural Dialog Generation
- Kaiser et al. [2018]: Fast Decoding in Sequence Models Using Discrete Latent Variables
- Sanchez et al. [2023]: Hidden Schema Networks

**Questions:**

1. The difference between Figure 3, 4 and 5 is the discrete bottleneck used, is it not? If so, the baseline results should always be the same across these figures, should they not? However, the results of the supervised baseline change in each figure. Does the supervised baseline also use the three different discrete bottlenecks? If so, what is the point of using the Gumbel DB in a supervised setting, if such a model is trained to maximise the likelihood of the target data wrt. the class probabilities ($\mathbf{s}$ in the author’s notation)?

2. How do the authors understand the difference in performance of the proposed model between the softmax and VQVAE instances (as e.g. in Fig 2)? What about the supervised model?

3. How long does it take to train the proposed model as compared with the baselines?

4. How important is the size of the non-parallel dataset? In other words, did the authors consider changing the size of the non-parallel dataset?

5. Is it possible to use back-translation in the experimental setting proposed by the authors. If so, why wasn’t it considered?

---

### Official Review · Reviewer_fHTj · 2024-11-03

**Soundness:** 3
**Presentation:** 2
**Contribution:** 2
**Rating:** 5
**Confidence:** 3

**Summary:**

In this paper the authors propose a method called $\Sigma$AE (symbolic auto-encoding) for mapping between two different symbolic system $X$ and $Z$ (sets of discrete sequences, e.g. natural language). It relies on two decoders $M_{xz}$, from $X$ to $Z$ and $M_{zx}$ from $Z$ to $X$ that output, via a bottleneck (DB) head, both a probability distribution over tokens $\mathbf{p}$ and a sample of this distribution $\mathbf{v}$ obtained via quantisation. The method works in two different modalities: unsupervised and weakly-supervised. During unsupervised training, one inputs points of $X$ ($Z$) to $M_{xz}$ ($M_{zx}$), whose output (sample) sequence is passed as input to $M_{zx}$ ($M_{xz}$) returning back to the input space and training by reconstruction. During weakly-supervised traning one alternates the unsupervised objective with supervised training via paired data, minimising the standard cross-entropy loss using the distribution outputs of the two decoders. The authors experiment with three different types of well-known quantisation mechanisms that enable back-propagating the gradients during the proposed unsupervised training: softmax-argmax, gumbel-softmax-argmax, vq-vae quantisation. Additionally, to alleviate a phenomenon called hidden sequence collapse noticed the authors (where the model considers only the first tokens of the bottleneck sequence are considered and the end-of-sequence token is selected prematurely), they introduce a straight-trough gradient estimation of the masking probabilities given the EOS token (i.e., pass the gradient of the distribution of the EOS masking instead of back-propagating only on the tokens before the EOS). The authors show results on 4 different seq2seq datasets: SCAN, PCFG SET, CFQ and COGS and compare with a fine-tuned T5 and a GPT3.5 model (in-context learning).

**Strengths:**

- The idea of mapping into an emergent language is interesting. I believe the main applications would be in settings where we need to translate non-symbolic domains to symbolic ones moreso than the presented symbolic to symbolic setting, which I understand is more easy to explore first (e.g., map an image to a a discrete sequence that represents its concept). This capacity of compressing data into a compact emergent language in an unsupervised way is still not available. VQ-VAE claims to do so but it does not do so in a natural way since it maps data in a local fashion, describing patches of images with the symbols, not the image itself. Humans instead describe information globally in a very compact way (“a red cat” should suffice to describe an image of a red cat). With images tough reconstion is ill-posed when latents are too compressed. This study, which assumes the same information content between $X$ and $Z$, is an initial foray into investigating more the idea of an emergent language bottleneck.
- The idea of doing a straight-through estimation of the gradients related to the EOS masking is novel to my knowledge, and it can be used in different scenarios. Still the authors do not show any quantitative evaluation of hidden sequence collapse (see weaknesses).

**Weaknesses:**

- The paper is difficult to follow and its structure could be improved in different ways:
   - First the authors do not explain in a clear way the main idea of their paper in the introduction. The authors only write (wrt. the unsupervised part): “$\Sigma AE$ performs two autoencoding tasks through discrete bottlenecks: one model encodes and reconstructs data from $X$ to $Z$, while the other does so from $Z$ to $X$.” This should be interpreted that you do x->z (as latent variable)-> x (back). But is not clear at all it should be like this here. While there is a Figure 1 describing the method, it is too cryptic to follow at that stage of the paper. I understood it after studying Section 3.1 which is in the Experiments section (on Page 8) , not even in the methodological section.
   - Different parts seem to be redundant / repetitive. Section 2.1.1 on REINFORCE / Annealing methods and Reparameterization trick should take only one or two lines since they are well known methods from literature and at least REINFORCE and Annealing methods are not used in the paper. Maybe the Gumbel-Softmax is ok to put since is directly used. Another example, why re-define the mask in Eq. 13 if it is already defined in previous sub-section?
   - There are sometimes symbols that are used without being defined and imprecisions on a notational level. For example in Line 360, there is $\mathbb{P}(O_k = \text{< EOS >})$, but $O$ is defined in the appendix. Or on Line 238 $\mathbf{s}$ was never introduced. I list other situations like this in the questions part.

- Experimental section is lacking.
    - While SCAN, PCFG SET, CFQ and COGS are valid benchmarks on compositional generalization, I do not understand why there isn’t any benchmark on natural language translation) (and relative evaluation on standard BLEU score),  which is the standard symbolic transduction task in NLP (especially since the authors talk about the Rosetta Stone analogy in Figure 1).
    - There isn’t any quantitative evaluation of hidden sequence collapse phenomenon.
    - The authors do not discuss which quantisation method does work better: for example VQ DB seems doing pretty bad in Table 1 especially on CFQ (0.00 sentence reconstruction) but is achieving the best results on the weakly supervised benchmark (Figure 2), especially on CFQ.

**Questions:**

Here I list different typos / question on the text.

- 21: Typo: Our after coma.
- 29: The authors do not give a definition of “symbolic system”. I assume it is a set of discrete sequences with a probability distribution on top.
- 41: “a self-supervised framework”: the method is not really self-supervised since it is not useful at all if paired data is not introduced. Authors should refer to it as a weakly-supervised framework.
- 44: Maybe more of a curiosity since notation style is at the discretion of the authors: why naming the two symbolic systems $X$ and $Z$ instead of $X$ and $Y$? To give the impression that the second variable $Z$ since it is output of $M_{xz}$ it can be seen as a latent variable? This would be ok if one wouldn’t have also $M_{zx}$. In my opinion using directly $X$ and $Y$ would have been more elegant since there isn’t a preferred latent variable in this case ($Z$ is for $X$ and $X$ is for $Z$).
- 45: The illustration has $D_{XZ}$ on top of the writing “Parallel Data”.
- 92: Fortuin et al. (2019) investigated training for what task?
- 97: What it means “which may depend on subsequent stages of the computation graph.”?
- 127: REINFORCE was published by Williams in 1992 not 2004.
- 136: The annealing methods are explained in an insufficient manner.
- 138: The expected value?
- 142: Why put $\theta$ both on subscript and as function argument? Ah… from the equation below I see $\theta$ should be $x$. Please correct.
- 184: “would introduce information leakage through the averaging weights from the encoder through the categorical distribution”. I do not understand this phrase? Could the authors explain better what they mean with it?
- 186: So $\mathbf{v}$ is a vector which contains the $z$s? If here we are explain a single variable why we stick with multiple variables before Section 2.3?
- 191: If $\mathbf{v}$ is a sequence then $\mathbf{p}$ is a matrix, not a vector.
- 203: “At its core” to “At their core”.
- 203: VQ-VAE does not estimate a probability. I see the authors later describe it as a degenerate delta distribution, but I think this phrase could be written better.
- 238: What is $\mathbf{s}$? is there a typo and the previous $\mathbf{p}$ should be $\mathbf{s}$?
- 340: The latent sequences instead of “the latent”.
- 360: The $O_k$ defined in appendix instead of main body.
- 368: As said in Weaknesses, I think Section 3.1 should be at the beginning of methods part instead of end of page 8.
- 374: Again calling $Z$ latent variable is confusing because at the beginning of text is seen as data variable.
- 425: Why notably? It is something straightforward.
- 440: “You can find” is not very formal.
Table 1: It is not true a high accuracy is achieved with all methods, e.g. VQ DB is very low on sentence accuracy on PCFG, COGS and CFQ.
- 474: “that where the latent is discrete and sequential” is written in a bad way.
- 613: I think is far stretched to say that this phenomenon is related to dropout.

---

### Official Review · Reviewer_88T4 · 2024-11-04

**Soundness:** 3
**Presentation:** 3
**Contribution:** 3
**Rating:** 6
**Confidence:** 3

**Summary:**

The paper presents Symbolic Autoencoding, a noval way to train sequence models using both paired and unpaired data by creating a symbolic language in between. The authors test their method on four different tasks and show it works better than existing approaches when there is limited paired data available. They use three different ways to estimate gradients to make their method work with discrete variables, though they note some limitations in training speed.

**Strengths:**

1. The proposed framework enables learning from both parallel and non-parallel data, addressing a common limitation in current sequence-to-sequence models that typically require large amounts of parallel data.
2. The authors thoroughly validate the approach by testing three different discrete bottleneck implementations, providing detailed comparisons and analyses across four different sequence transduction tasks, with strong empirical results against multiple baselines.
3. The authors shows particularly strong performance in low-resource scenarios, which is a common real-world challenge.

**Weaknesses:**

1. The reliance on synthetic datasets, which limits applicability to real-world language.
2. The randomness by the Gumbel-Softmax discretizer reduces sentence-level accuracy consistency, impacting the model’s reliability in token-precise tasks.

**Questions:**

I have some questions here:
1. How does the framework handle more complex, real-world language data with high semantic variability?
2. Could the authors clarify the practical impact of the Gumbel-Softmax noise on downstream applications? Specifically, are there scenarios where this instability could significantly hinder model performance?
3. The paper mentions that unsupervised data contributes less to accuracy gains; have you considered alternative strategies to better leverage non-parallel data in low-resource settings?

---

### Official Review · Reviewer_vYPY · 2024-11-05

**Soundness:** 1
**Presentation:** 1
**Contribution:** 1
**Rating:** 3
**Confidence:** 4

**Summary:**

In this paper, the authors describe a sequence autoencoder that has a discrete sequence bottleneck (middle layer). To train this model efficiently and effectively, the authors describe a few different approaches to smooth out the discretization happening in this bottleneck layer; vector quantization as in VQ-VAE and a couple of variants of straight-through estimators. The authors suggest that such a sequence autoencoder is able to learn a meaningful symbolic representation of the input sequence and enables seamless integration of both labelled and unlabelled sequences.

**Strengths:**

Unfortunately I could not find much of strengths in this manuscript, since most (if not all) of the ideas in this paper have already been studied extensively over the past decade. Although I welcome the revival of discrete bottleneck based approaches to sequence modeling personally, I could not tell exactly what additional insights, we as the community hadn’t had about such approaches, are conveyed in this study.

**Weaknesses:**

Based on my reading of the manuscript, the authors may have missed quite a few earlier studies that effectively focused on the same problem. For instance, Miao & Blunsom (2016; https://arxiv.org/abs/1609.07317) proposed a discrete sequence bottleneck model for sentence compression, where they used reinforcement learning and semi-supervised learning. Learning a dynamic-length latent sequence was studied earlier in many different settings, and i can think of e.g. Evtimova et al. (2017; https://arxiv.org/abs/1705.10369) and Graves (2016; https://arxiv.org/abs/1603.08983). Of course these earlier studies do not follow the exact procedures in this manuscript, but these differences seem relatively minor (e.g. reparametrized gradient estimator vs. log-derivative trick, etc.) The authors should look at these papers as well as any follow-up studies to these and reposition their work more carefully.

One of the main promises by the authors already in the abstract is that their approach results in better “interpretability and structure”. Unfortunately, I cannot see why this is the case and do not see any evidence behind this. The authors need to give more thoughts on how they define interpretability, why such interpretability is desirable, how their approach improves interpretability defined in this way and how they can show this is the case. The same applies to “structure”.

**Questions:**

Please see my answer above.

---

### Author Response · Authors · 2024-11-28
**Summary of the changes in the updated draft**

We have carefully addressed the reviewers comments with the key changes outlined below. These changes have been highlighted in the updated draft for your convenience.

- Training Time: Added a discussion comparing training times for unsupervised and supervised steps, expanded the explanation of the ICL baseline, and detailed dataset choices (A3, A6).

- Related Work: Included missing references and a discussion on reinforcement learning methods for symbolic representation learning.

- Methodology: Enhanced the introduction and conclusion with a clearer and more detailed overview of the methodology.

- Notation Fixes: Resolved inconsistencies in the probabilistic model in Sections A.2 and 2.2.1.

- Sequence Collapse: Expanded discussion on the sequence collapse problem (Sections 2.3.1 and A.2).

- Annealing Methods: Provided more detail on continuous averaging and annealing methods (Section 2.1.1).

- Text Improvements: Addressed clarity and consistency issues as suggested by reviewers.

We hope these revisions address your concerns and improve the manuscript. Thank you for your valuable input.

---

### Meta-Review · Area_Chair_yb3E · 2024-12-20

**Metareview:**

The paper proposes Symbolic Autoencoding (ΣAE), a novel framework for sequence transduction. The approach demonstrates strong performance in low-resource settings. Evaluations on various datasets highlight the framework’s ability to compress data into symbolic representations and reconstruct it. The authors emphasize ΣAE’s potential for interpretability and compositional generalization.

The paper presents a novel approach by using STGEs to train symbolic discrete latent spaces and addresses the challenge of low-resource settings, which is important in sequence transduction tasks.  However, the paper lacks the evaluation on real-world tasks, such as machine translation or semantic parsing, which raises questions about the general applicability of the method. Key comparisons with standard baselines, like back-translation or other symbolic autoencoders, are missing. The results are not always convincing, particularly at higher supervision levels, and the analysis of supervision ratios and dataset complexity is insufficient. The high computational cost of training limits the scalability of the approach.

With these concerns,  I recommend rejection for this submission. I hope the authors can improve the paper and submit it to a future venue.

**Additional Comments On Reviewer Discussion:**

## Points Raised by Reviewers:
1. The reviewers questioned the lack of real-world tasks and standard baselines (e.g., back-translation, existing symbolic autoencoders).
2. The results were deemed unconvincing, especially in higher supervision regimes where baselines like T5 performed comparably or better.
3. The mathematical notation and presentation were criticized for being unclear and inconsistent.
4. Reviewers requested more information on training times and an analysis of supervision ratios across datasets.
5. Some reviewers noted the lack of a quantitative evaluation of the "hidden sequence collapse" phenomenon described in the paper.

## Author Responses:
1. The authors clarified that the choice of synthetic datasets was intentional to evaluate compositional generalization. They acknowledged the absence of real-world tasks and back-translation baselines but argued that these were outside the scope of the current work.
2. They provided additional details about training dynamics and clarified the differences between bottleneck mechanisms (e.g., Gumbel vs. Softmax).
3. The authors revised the manuscript to address notational inconsistencies and improve the presentation of results.
4. They discussed the computational trade-offs of their approach and highlighted its relevance for low-resource settings.

## Weighting Points in the Final Decision:
While the authors addressed some concerns (e.g., clarity and notational issues), the lack of real-world benchmarks and insufficient baseline comparisons remained significant drawbacks. The reviewers appreciated the conceptual novelty of the framework but found the experimental evaluation insufficient to establish its broader applicability and soundness. The high computational complexity further limits the feasibility of scaling the approach to practical tasks.

---

### Decision · Program_Chairs · 2025-01-22

Reject